# Breaking the Language Barrier:
# Improving Cross-Lingual Reasoning with Structured Self-Attention

**Negar Foroutan**[*], **Mohammadreza Banaei**[*], **Karl Aberer, Antoine Bosselut**
EPFL
{negar.foroutan,mohammadreza.banaei,antoine.bosselut}@epfl.ch

## Abstract

In this work, we study whether multilingual language models (MultiLMs) can transfer logical reasoning abilities to other languages when they are fine-tuned for reasoning in a different language. We evaluate the cross-lingual reasoning abilities of MultiLMs in two schemes: (1) where the language of the context and the question remain the same in the new languages that are tested (*i.e.*, the reasoning is still monolingual, but the model must transfer the learned reasoning ability across languages), and (2) where the language of the context and the question is different (which we term code-switched reasoning). On two logical reasoning datasets, RuleTaker and LeapOfThought, we demonstrate that although MultiLMs can transfer reasoning ability across languages in a monolingual setting, they struggle to transfer reasoning abilities in a code-switched setting. Following this observation, we propose a novel attention mechanism that uses a dedicated set of parameters to encourage cross-lingual attention in code-switched sequences, which improves the reasoning performance by up to 14% and 4% on the RuleTaker and LeapOfThought datasets, respectively.[1]

## 1 Introduction

Recent studies show that language models (LMs) are capable of logically reasoning over natural language statements (Clark et al., 2020b), reasoning with their implicit knowledge (Talmor et al., 2020), and performing multi-step reasoning via chain-of-thought prompting when the model size is large enough (Wei et al., 2022b; Kojima et al., 2022; Wei et al., 2022a). A separate line of work has focused on pre-training language models on multilingual corpora to enable knowledge transfer across different languages. These efforts led to multilingual

Figure 1: An example of monolingual and code-switched reasoning. In code-switched reasoning, the context and question are in different languages.

language models (MultiLM) such as mBERT (Devlin et al., 2019), mT5 (Xue et al., 2021), and XLM-R (Conneau et al., 2020), which have been shown to generalize in a zero-shot cross-lingual setting (Pires et al., 2019a; Conneau and Lample, 2019). The cross-lingual transfer is often enabled by fine-tuning the MultiLM on a high-resource language (typically English) and then evaluating it on other target languages.

However, as most of the recent efforts on reasoning-related tasks have been centered around English, our knowledge of the multilingual reasoning capabilities of language models remains limited. In this work, we investigate the logical reasoning capabilities of MultiLMs, especially in monolingual and *structured* code-switched[2] settings (Figure 1). In the monolingual setting, the context and the question are in the same language. In the *structured* code-switched setting, we refer to a setting where the context and question are in two different languages. The code-switched setting can be found in many realistic scenarios, such as when non-English speakers may ask questions about information that is unavailable in their native language (Asai et al., 2021).

For both reasoning settings, we conduct experiments using the RuleTaker dataset (Clark et al., 2020b), which contains artificial facts and rules, and the LeapOfThought dataset (Talmor et al., 2020), which incorporates real-world knowledge

---

[*]Equal contribution
[1]Our code is available at https://github.com/negar-foroutan/multilingual-code-switched-reasoning.

[2]Throughout the paper, we will use the terms "structured code-switching" and "code-switching" interchangeably.

into the reasoning context. Our results show that although MultiLMs perform well when fine-tuned in different languages (*i.e.*, high *in-language* performance when fine-tuning and testing on the same language), their cross-lingual transfer can vary considerably, especially in the code-switched setting. We posit that the lack of code-switched data in MultiLM pre-training data makes fine-tuning on code-switched data inconsistent with pre-training.

To improve the code-switched reasoning capabilities of MultiLMs, we propose two methods. First, we propose a dedicated *cross-lingual query* matrix (section 4.1) to better model cross-lingual attention when the MultiLMs receive code-switched sequences as input. This query matrix is pre-trained on unsupervised code-switched data, either shared across all language pairs or specific to a single one. Then, we propose a structured attention dropout (see section 4.1), in which we randomly mask attention between tokens from different languages (*i.e.*, context-question attentions) during training. This masking makes the fine-tuning phase more consistent with the pre-training by regularizing cross-lingual attention.

By mixing the two methods, we also experiment with an *interfered* variant of the cross-lingual query, which considerably improves cross-lingual generalization, especially in code-switched settings. We evaluate our methods for the code-switched setting and show they improve the cross-lingual transfer of MultiLMs by 14% and 4% for the RuleTaker and LeapOfThought datasets, respectively.

## 2 Motivation

Most prior work on reasoning with language models remains limited to monolingual (English) systems (Han et al., 2021; Sanyal et al., 2022; Shi et al., 2023; Tang et al., 2023). In this work, we investigate the reasoning abilities of MultiLMs, formulating an analysis in *formal reasoning* that evaluates MultiLMs on their ability to resolve logical statements. Given a set of facts and rules as *context* (in natural language sentences), the task is to predict whether a given *statement* is true.

In our multilingual reasoning setting, we assume a given set of languages $= \{L_1, L_2, ..., L_N\}$, and define $L_c$ and $L_q$ as the context and statement languages, respectively. Typically, MultiLMs are evaluated in a monolingual setup where $L_c = L_q$. However, if MultiLMs are truly multilingual, we posit that they should also be able to reason in a scenario

where $L_c \neq L_q$. Thus, to evaluate the multilingual reasoning ability of MultiLMs, we first define four different evaluation setups based on the language of context or statement: (1) both the context and statement are always in one language (monolingual reasoning); (2) the context is always in one language, and the statement can be in any language; (3) the context can be in any language, but the statement is always in one language; and (4) both the context and statement can be in any language.

To have a reasonable baseline to compare with the code-switched setups, we first focus on the monolingual evaluation (1), in which we evaluate the reasoning ability of MultiLMs for nine typologically different languages. Then, by fine-tuning the models on code-switched data, we evaluate their performance for setups (2) and (3) where either the language of the context or the language of the question is different from the training data. This evaluation aims to study the possibility of teaching models to reason across languages in a code-switched setting, and to investigate the extent they can transfer their reasoning to other code-switched data formats. Finally, we hypothesize that in order to succeed in setup (4), the model would have to be strong in setups (2) and (3). Since our experimental results show that the MultiLMs struggle in these two setups, we focus on improving their performance for setups (2) and (3).

## 3 Multilingual Reasoning

In this section, we describe our evaluation of the logical reasoning capabilities of MultiLMs for monolingual and code-switched settings.

### 3.1 Analysis Setup

We run our experiments on two datasets focusing on multi-hop logical reasoning over natural language knowledge:

**RuleTaker.** This is a set of five datasets, each constrained by the maximum depth of inference required to prove the facts used in its questions (Clark et al., 2020b). This dataset is generated with the closed-world assumption (CWA), assuming a statement is false if it is not provable. Each example consists of facts and rules (*i.e.*, context) and a statement (more details in Appendix A.1).

**LeapOfThought (LoT).** This dataset comprises ∼30K true or false hypernymy inferences, verbalized using manually written templates (Talmor

et al., 2020). The hypernymy relations and properties are derived from WORDNET (Fellbaum, 1998) and CONCEPTNET (Speer et al., 2017). This dataset contains two main test sets; EXPLICIT REASONING which performs inference over explicit natural language statements, and IMPLICIT REASONING where the model must reason by combining the context with missing information that should be implicitly encoded by the model. We create a modified version of LoT, and use the IMPLICIT REASONING test set in our evaluation. The dataset modification pipeline and the reason behind using only the IMPLICIT evaluation setting is further discussed in Appendix A.2.

**Models.** We conduct all our experiments using the cased version of multilingual BERT (mBERT; Devlin et al. 2019) and the base version of XLM-R (Conneau et al., 2020). We train a binary classifier on top of the model's classification token (*e.g.*, *[CLS]* in mBERT) to predict whether a given statement/question is true or false. The model's input is *[CLS] context [SEP] statement [SEP]* and the *[CLS]* output token is used for the classification. For evaluation, we measure the model's accuracy. We use full fine-tuning for these experiments. The random baseline is 50% (binary classification).

**Languages.** Both RuleTaker and LoT datasets are only available in English. We translated these two datasets into eight languages using the Google Translate API. We have chosen typologically diverse languages covering different language families: Germanic, Romance, Indo-Aryan, and Semitic, and including both high- and medium-resource languages from the NLP perspective. These languages include French (fr), German (de), Chinese (zh), Russian (ru), Spanish (es), Farsi (fa), Italian (it), and Arabic (ar).

### 3.2 Reasoning Over Monolingual Data

The average in-language and cross-lingual zero-shot performance of mBERT for each source language are depicted in Table 1. For the cross-lingual zero-shot performance, we first fine-tune models on a single source language, test it on other languages, and then take an average of these results.

On the RuleTaker dataset, the model is able to learn the task for the Depth-0 subset nearly perfectly for almost all the languages, exhibiting relatively high cross-lingual transfer performance (∼87%). However, for models trained on higher depths (*i.e.*, requiring more reasoning

hops), the model's performance drops for both in-language and cross-lingual evaluation settings, and the performance gap between different source languages increases. Moreover, when increasing the depth, zero-shot cross-lingual performance suffers more compared to in-language performance, showing that as the complexity of the task increases, the harder it becomes to generalize to other languages.

For the LoT dataset, the model must learn to reason by combining its implicit knowledge of hypernyms with the given explicit knowledge. However, the model's performance differs for different languages, suggesting that the model's ability to access and use the implicit knowledge is not the same for all languages. We also observe that a language with high in-language performance does not necessarily have a high zero-shot cross-lingual performance. We hypothesize that for some languages, the model starts learning in-language noises that are not generalizable to other languages.

We generally observe the same patterns for the XLM-R model (see Appendix B) when fine-tuned on the monolingual RuleTaker and LoT datasets.

### 3.3 Reasoning Over Code-Switch Data

When we fine-tune the model using a code-switched data format, the context is in one language and the statement is in another. In our experiments, we use English as an anchor language for the context (*i.e.*, en-X) or for the statement (*i.e.*, X-en). In the fine-tuning phase, we learn the task using the en-X data format, and evaluate it on both en-X and X-en data formats. The models' average in-language and zero-shot cross-lingual performance are shown in Table 2.

For Depth-0 of the RuleTaker dataset, mBERT is able to learn the reasoning task almost perfectly for most languages. As the depth of the task increases, the performance of the code-switched reasoning declines. This decline is more pronounced at higher depths compared to the monolingual scenario. While the model is capable of learning reasoning within this framework, its zero-shot generalization to other code-switched data, such as en-X (where the context language remains English but the statement language changes), is poor. Reasoning over two languages poses a greater challenge than reasoning within monolingual data due to the need for information alignment across languages. Consequently, the transferability of such tasks to other language pairs becomes more challenging.

| | RuleTaker | | | | | | | | LeapOfThought | |
|---|---|---|---|---|---|---|---|---|---|---|
| | Depth-0 | | Depth-1 | | Depth-2 | | Depth-3 | | | |
| | in-lang. | cross-ling. | in-lang. | cross-ling. | in-lang. | cross-ling. | in-lang. | cross-ling. | in-lang. | cross-ling. |
| **en** | 100.00 | 87.96 | 93.37 | 73.60 | 88.00 | 67.91 | 88.46 | 67.13 | 81.15 | 62.11 |
| **fr** | 99.40 | 87.06 | 90.50 | 74.82 | 86.64 | 65.45 | 83.70 | 67.55 | 80.78 | 65.12 |
| **fa** | 99.99 | 87.39 | 90.04 | 67.81 | 86.96 | 63.71 | 84.64 | 63.53 | 66.39 | 64.37 |
| **de** | 99.41 | 89.57 | 90.77 | 76.67 | 85.41 | 71.57 | 83.10 | 70.74 | 77.11 | 67.03 |
| **ar** | 99.48 | 80.20 | 90.35 | 72.32 | 84.81 | 67.79 | 82.62 | 62.21 | 69.62 | 67.71 |
| **es** | 99.99 | 89.68 | 91.84 | 76.20 | 88.16 | 72.29 | 85.79 | 68.75 | 75.25 | 64.22 |
| **zh** | 100.00 | 87.48 | 92.43 | 72.46 | 89.04 | 68.13 | 85.94 | 66.25 | 84.12 | 62.32 |
| **ru** | 99.97 | 89.61 | 90.54 | 78.05 | 86.43 | 70.88 | 84.01 | 67.08 | 70.60 | 68.02 |
| **it** | 99.81 | 90.09 | 93.14 | 78.28 | 86.95 | 74.01 | 84.64 | 70.43 | 74.99 | 64.68 |
| **Average** | 99.78 | 87.67 | 91.44 | 74.47 | 86.93 | 69.08 | 84.77 | 67.07 | 75.56 | 65.06 |

Table 1: **Monolingual Setting:** In-language and cross-lingual zero-shot performance (accuracy) of the mBERT model for the RuleTaker and LeapOfThought datasets. Cross-lingual performance is the average performance of the model being fine-tuned on a single source language and then zero-shot transferred to other languages.

| | RuleTaker | | | | | | | | | | | | LeapOfThought | | |
|---|---|---|---|---|---|---|---|---|---|---|---|---|---|---|---|
| | Depth-0 | | | Depth-1 | | | Depth-2 | | | Depth-3 | | | | | |
| | in-lang. | en-X | X-en | in-lang. | en-X | X-en | in-lang. | en-X | X-en | in-lang. | en-X | X-en | in-lang. | en-X | X-en |
| **en-fr** | 99.29 | 54.82 | 53.47 | 93.34 | 55.28 | 51.85 | 87.78 | 54.78 | 51.83 | 83.26 | 54.14 | 50.28 | 79.57 | 73.47 | 71.48 |
| **en-fa** | 97.46 | 54.04 | 52.05 | 87.72 | 62.17 | 51.56 | 70.95 | 53.64 | 51.29 | 62.26 | 50.95 | 50.52 | 74.99 | 73.82 | 65.93 |
| **en-de** | 99.63 | 54.26 | 52.69 | 88.85 | 52.67 | 51.87 | 83.97 | 55.32 | 52.73 | 79.08 | 53.48 | 51.61 | 77.60 | 71.16 | 65.09 |
| **en-ar** | 85.93 | 53.73 | 52.36 | 67.05 | 57.83 | 51.92 | 68.54 | 55.33 | 51.74 | 61.29 | 52.78 | 50.76 | 77.09 | 75.35 | 64.57 |
| **en-es** | 99.99 | 57.25 | 56.29 | 90.18 | 54.34 | 50.91 | 86.54 | 58.20 | 53.15 | 78.09 | 55.53 | 51.72 | 79.29 | 75.62 | 72.55 |
| **en-zh** | 100.00 | 52.68 | 51.81 | 92.34 | 54.70 | 51.31 | 81.41 | 54.92 | 51.24 | 68.74 | 52.83 | 50.00 | 84.85 | 68.92 | 61.09 |
| **en-ru** | 98.03 | 58.40 | 52.06 | 94.02 | 59.70 | 50.64 | 80.28 | 57.69 | 51.96 | 73.89 | 56.26 | 50.87 | 76.57 | 74.64 | 65.11 |
| **en-it** | 99.91 | 56.26 | 54.68 | 92.25 | 52.88 | 50.94 | 85.59 | 54.58 | 51.20 | 79.50 | 53.29 | 51.11 | 75.38 | 70.53 | 66.90 |
| **Average** | 97.53 | 55.18 | 53.18 | 88.22 | 56.20 | 51.38 | 80.63 | 55.56 | 51.89 | 73.26 | 53.66 | 50.86 | 78.17 | 72.94 | 66.59 |

Table 2: **Code-Switched Setting:** In-language and cross-lingual zero-shot performance (accuracy) of the mBERT model for the RuleTaker and LeapOfThought datasets. In-language performance corresponds to evaluating the model in the same language as the training data.

On the LoT dataset, the model performs quite well on the code-switched data, outperforming the monolingual scenario for nearly all languages. The relatively high code-switched performance shows that the language of the context plays an important role in accessing the implicit knowledge encoded in the model's parameters, as the model must rely on this knowledge to solve the task. Providing the context in English facilitates access to implicit knowledge compared to other languages. This is also inline with the empirical observation that generalization to X-en is considerably worse than en-X. We generally observe the same pattern for the XLM-R (see Appendix B) when fine-tuned on the monolingual RuleTaker and LoT datasets.

Following the empirical observations showing MultiLMs struggle to transfer reasoning abilities in a code-switched setting, we propose a novel attention mechanism to mitigate the lack of code-switched transfer in these models.

## 4 Cross-lingual-aware Self-Attention

Although MultiLMs have been pre-trained on multilingual corpora, individual inputs to the model stay mostly monolingual (Devlin et al., 2019; Conneau et al., 2020). When these models are fine-tuned on a code-switched downstream task, unlike the pre-training phase, tokens from different languages can attend to each other, which, as demonstrated in Tables 2 and 8, results in poor generalization to other code-switched language pairs. We also observe that self-attention patterns considerably change when we compare code-switched in-language and cross-lingual samples' attention patterns[3] (see Figure 4).

### 4.1 Approach

In order to make the fine-tuning phase more consistent with the pre-training, we propose two sets of methods to better handle the cross-lingual interactions of tokens.

**Cross-lingual Query** To better model the cross-lingual attention for code-switched tasks, we pre-train a *cross-lingual* query matrix $Q_{cross}$ (while keeping all other parameters frozen) on code-switched unsupervised data (more experimental

---

[3]The two samples are semantically the same, only having different statement languages.

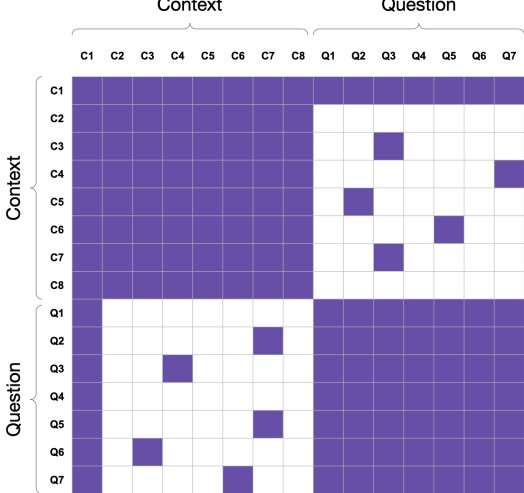

Figure 2: Illustration of the drop attention scheme. Due to the input's code-switched structure, we want to limit the attention between context and question tokens. It can be seen that tokens from the same language can fully attend to each other, but there is a dropout (white cells) when cross-lingual attention is being applied. In order to ensure a reliable *bridge* between context and question, the first token (*e.g.*, [CLS] in mBERT) attends to all tokens, and also all tokens attend to the first token.

details in section 4.2). More specifically, we use two sets of attention masks, $M_1$ and $M_2$, where $M_1$ enforces the query matrix $Q$ to focus only on monolingual attentions, and $M_2$ constrains the cross-lingual query $Q_{cross}$ to cross-lingual attentions (see Figure 3.a). Formally, the self-attention probabilities for a given attention head, up to a (row-wise) normalization term, are computed as below:

$$M_1 \odot exp(\frac{QK^T}{\sqrt{d}}) + M_2 \odot exp(\frac{Q_{cross}K^T}{\sqrt{d}})$$

where $Q$ and $K$ are the query and key matrices, $d$ is the model's hidden dimension, and $\odot$ represents the Hadamard product. It is worth noting that this scheme still allows attention between all tokens; however, monolingual and cross-lingual attentions are handled by different query matrices.

The proposed $Q_{cross}$ can either be pre-trained for a single language pair (*e.g.*, en-fr pair where context is in English and question/statement is in French), or it can be shared across many language pairs. We show in Section 4.3 that having language-pair specific $Q_{cross}$ enables *modularity*, meaning a model that is trained on a given *source* language pair can perform considerably better on another language pair by just swapping the source $Q_{cross}$ matrices with the target ones.

**Structured Attention Dropout**   As mentioned earlier, poor generalization of MultiLMs in code-switched settings can be attributed to inconsistency between the pre-training and fine-tuning phases, where the former mostly deals with monolingual attention while the latter needs to handle cross-lingual attention as well. We propose that the consistency can be improved by limiting the cross-lingual attention in the fine-tuning phase (*i.e.*, regularizing computational interactions between languages). As demonstrated in Figure 2, this can be achieved by **randomly masking** attention scores (*i.e.*, attention dropout), with probability $P_{mask}$, when tokens from different languages attend to each other. Moreover, to ensure a reliable *bridge* between context and question, we never mask the attention scores of the first token (*e.g.*, [CLS] in mBERT) to help the model better flow information between two sections. Table 11 demonstrates the importance of *structured* attention dropout for better generalization in code-switched settings.

**Interfering Cross-lingual Query**   Given the promising performance of the attention dropout for code-switched tasks, we experiment with a variation of cross-lingual query, where queries Q and $Q_{cross}$ also partially handle cross-lingual and monolingual attentions, respectively (see Figure 3b). We empirically observe that having attention masks that could randomly *interfere* with each other generally results in better performance (see Table 12) compared to the attention masks proposed in Figure 3a. In this scheme, $M_1$ and $M_2$ are generated **randomly** and **online**,[4] but once sampled, the same masks will be used for all the attention heads in all layers (more details in Appendix D). Due to better empirical performance, this variation of the cross-lingual query will be used for all the following experiments.

## 4.2   Experimental Setup

All models are trained with the AdamW optimizer (Loshchilov and Hutter, 2017) using the HuggingFace Transformers (Wolf et al., 2020) implementation for Transformer-based (Vaswani et al., 2017) models. The hyperparameters used for performing different experiments can be found in Appendix C. All the reported scores are averaged over three different seeds.

---

[4]A given sample can have different attention masks in different epochs.

**Fine-tuning Setup.** As Bitfit fine-tuning outperforms full fine-tuning for all our experiments, we only report the Bitfit results here (Zaken et al., 2021). In Bitfit tuning, only biases are tuned in the MultiLM encoder, together with classifier and pooler parameters.

**Language Pairs.** To show the effectiveness of the proposed method, we fine-tune the models on four typologically diverse languages (language of the statement), namely fr, de, zh, and ru. Our analysis shows that combining monolingual and code-switched data in the fine-tuning step improves the reasoning performance. Moreover, a multilingual reasoner should be able to reason over both monolingual and code-switched data. So, for this set of evaluations, we use a combination of English and en-X (half of each) as the training dataset, which we denote mix(en, en-X).

**Pre-training Cross-lingual Query.** We train a *shared* (Shared $Q_{cross}$) or *language-pair specific* (Pair $Q_{cross}$) cross-lingual query matrix. For Shared $Q_{cross}$, a shared cross-lingual query is trained on a parallel code-switched variant of XNLI (Conneau et al., 2018a), where an English premise is followed by the same premise but in another language. For Pair $Q_{cross}$, we train a cross-lingual query for each en-X language pair again using the XNLI dataset. In both cases, only the cross-lingual query matrix is trained and the rest of the parameters are frozen. The training happens for 500K iterations.

**Baselines.** We compare the performance of the proposed method against two baselines: (1) The pre-trained model (**original**) (2) a model pre-trained on code-switched data (**CS-baseline**). For the CS-baseline, we pre-train the model on the parallel code-switched variant of XNLI (similar to the data we use to learn the shared cross-lingual query matrix) for 500K iterations to adapt the model to the code-switched setting.

**Cross-lingual Evaluation.** For all the experiments, we evaluate the zero-shot performance of the model on (1) a monolingual setting (where both context and question are in one language), (2) an en-X code-switched setting (where the context is in English and the question is in other languages), and (3) a X-en code-switched setting (where the question is in English and the context is in other languages). For the case when we Bitfit fine-tune the

model using a language-specific query matrix (Pair $Q_{cross}$), we use the query matrix of the target language during the inference (only the weights). For example, while doing the zero-shot evaluation on en-zh, we use the en-zh cross-lingual query matrix instead of the one from the fine-tuned model.

### 4.3 Experimental Results

Table 3 shows the average zero-shot transfer performance (accuracy) for the RuleTaker dataset. For both mBERT and XLM-R, introducing a shared cross-lingual query matrix (Shared $Q_{cross}$) improves the reasoning accuracy. These results underscore the significance of maintaining consistency between the pre-training and fine-tuning phases in code-switched downstream tasks to facilitate effective transfer learning.

Using a specific query matrix for each language pair (Pair $Q_{cross}$) further boosts the cross-lingual transfer performance across most tested settings (up to 18%). In this scenario, there is a dedicated set of parameters to learn the attention patterns for a language pair rather than having them share the same number of parameters among many different language pairs. In other words, dedicated parameters help the model learn attention patterns for specific language pairs.[5]

Interestingly, in many cases, our approach also improves the transfer performance for monolingual data (**mono**). We hypothesize that, by having a separate cross-lingual query matrix, the model does not need to learn the cross-lingual attention pattern using the same parameters, reducing the chance of overfitting to the code-switched format.

We also conducted a comparison with a code-switched baseline in which the MultiLM is pre-trained on a code-switched version of XNLI. The code-switched baseline (**CS-baseline**) showed improved transfer results for en-X format and, in some cases, performed competitively with the Pair $Q_{cross}$ approach. However, it negatively affected performance in monolingual and X-en scenarios, particularly for the mBERT model. In essence, the model exhibited overfitting to the language pairs in en-X format it was trained on, making it unable to generalize effectively to monolingual and other code-switched formats. On the other hand, both Shared $Q_{cross}$ and Pair $Q_{cross}$ demonstrated the ability to generalize their reasoning to

---

[5]There is no Pair $Q_{cross}$ for en-fa and en-it (as they are not part of the XNLI dataset), and all the transfer results for these two languages are fully zero-shot.

| Train Data | Method | Depth-0 | | | Depth-1 | | | Depth-2 | | | Depth-3 | | |
|---|---|---|---|---|---|---|---|---|---|---|---|---|---|
| | | **mBERT** | | | | | | | | | | | |
| | | mono | en-X | X-en | mono | en-X | X-en | mono | en-X | X-en | mono | en-X | X-en |
| mix(en, en-fr) | Original | 89.14 | 65.38 | 60.81 | 70.76 | 60.48 | 58.16 | 67.43 | 62.14 | 55.55 | 62.48 | 57.94 | 51.04 |
| | CS-baseline | 78.93 | 74.72 | 54.98 | 67.59 | 68.25 | 53.90 | 63.16 | 67.50 | 52.60 | 62.57 | 66.89 | 50.95 |
| | Shared $Q_{cross}$ | 92.52 | 70.07 | 65.16 | **77.72** | 67.26 | **63.93** | **74.81** | 64.23 | 58.97 | 70.46 | 63.86 | 55.75 |
| | Pair $Q_{cross}$ | **93.65** | **77.79** | **68.27** | 77.44 | **68.55** | 63.76 | 73.78 | **68.23** | **61.27** | **71.39** | **67.70** | **60.12** |
| mix(en, en-de) | Original | 88.71 | 66.75 | 59.10 | 68.98 | 58.64 | 56.69 | 73.39 | 62.88 | 55.66 | 63.45 | 57.36 | 50.84 |
| | CS-baseline | 84.77 | 74.73 | 57.06 | 68.08 | 67.88 | 53.99 | 63.58 | 67.47 | 52.42 | 62.23 | 66.18 | 50.73 |
| | Shared $Q_{cross}$ | 91.39 | 70.10 | 64.78 | 76.74 | 65.88 | 61.64 | 71.82 | 64.38 | 59.92 | **71.98** | 62.21 | 57.26 |
| | Pair $Q_{cross}$ | **94.11** | **76.32** | **69.85** | **77.38** | **68.31** | **63.22** | **73.79** | **68.42** | **62.16** | 70.56 | **67.23** | **61.86** |
| mix(en, en-ru) | Original | 91.69 | 69.25 | 60.23 | 76.49 | 60.40 | 57.09 | 68.99 | 57.62 | 52.93 | 65.60 | 57.70 | 50.05 |
| | CS-baseline | 83.65 | 75.92 | 54.68 | 71.06 | 69.96 | 55.49 | 64.80 | 66.84 | 52.47 | 60.06 | 58.93 | 48.71 |
| | Shared $Q_{cross}$ | **93.22** | 76.22 | 70.35 | **79.80** | 68.77 | **65.44** | 74.06 | 65.68 | 59.14 | **71.89** | 63.50 | 57.19 |
| | Pair $Q_{cross}$ | 92.23 | **77.22** | **71.87** | 78.31 | **74.00** | 64.50 | **74.67** | **67.97** | **63.47** | 70.98 | **66.73** | **60.10** |
| mix(en, en-zh) | Original | 91.20 | 65.58 | 59.80 | 76.43 | 63.02 | 57.20 | 68.23 | 55.40 | 52.47 | 65.03 | 56.55 | 50.85 |
| | CS-baseline | 83.16 | 70.22 | 57.46 | 67.34 | 66.87 | 54.29 | 66.29 | 65.73 | 53.01 | 60.72 | 63.64 | 52.21 |
| | Shared $Q_{cross}$ | **93.70** | 68.49 | 64.59 | 75.11 | 65.11 | 62.00 | 73.42 | 62.66 | 58.03 | 69.98 | 62.01 | 57.62 |
| | Pair $Q_{cross}$ | 93.21 | **72.09** | **69.83** | **78.93** | **67.12** | **64.22** | **75.82** | **66.65** | **60.52** | **71.34** | **66.35** | **60.05** |
| | | **XLM-R** | | | | | | | | | | | |
| mix(en, en-fr) | Original | 95.39 | 69.43 | 64.09 | 79.85 | 65.35 | 59.55 | 76.34 | 62.94 | 58.89 | 74.71 | 62.68 | 55.84 |
| | CS-baseline | 94.89 | 71.03 | 61.41 | 81.11 | 67.08 | 57.16 | 75.78 | 65.32 | 52.33 | 72.28 | 63.77 | 51.16 |
| | Shared $Q_{cross}$ | 95.92 | 74.78 | 70.87 | 79.82 | 68.46 | 63.84 | 79.99 | 70.64 | 62.14 | **77.26** | 68.55 | **60.81** |
| | Pair $Q_{cross}$ | **95.94** | **78.36** | **75.80** | **83.64** | **70.17** | **63.94** | **81.39** | **71.59** | **64.37** | 76.03 | **69.04** | 60.12 |
| mix(en, en-de) | Original | 94.95 | 65.72 | 64.94 | 82.58 | 64.99 | 62.03 | 78.74 | 63.88 | 57.02 | 75.06 | 64.87 | 58.02 |
| | CS-baseline | 91.92 | 72.53 | 57.14 | 76.70 | 66.64 | 54.29 | 73.25 | 65.58 | 52.20 | 74.78 | 62.87 | 51.87 |
| | Shared $Q_{cross}$ | **96.23** | 71.29 | **70.95** | 81.95 | 67.25 | 64.27 | **82.14** | 70.48 | 63.28 | 75.26 | 67.16 | 57.01 |
| | Pair $Q_{cross}$ | 96.19 | **73.61** | 70.89 | **84.33** | **68.40** | **65.23** | 80.11 | **71.72** | **64.55** | **76.73** | **69.89** | **59.78** |
| mix(en, en-ru) | Original | 94.46 | 72.86 | 63.94 | 80.80 | 66.55 | 59.53 | 78.23 | 65.90 | 55.78 | 74.33 | 63.05 | 53.32 |
| | CS-baseline | 95.02 | 74.63 | 60.42 | 80.96 | 71.53 | 54.85 | 78.30 | 67.56 | 52.84 | 68.59 | 64.93 | 49.43 |
| | Shared $Q_{cross}$ | **95.43** | 80.20 | 77.23 | 83.73 | 72.16 | **68.02** | **81.39** | 71.31 | **63.25** | 75.60 | **69.17** | 56.23 |
| | Pair $Q_{cross}$ | 95.14 | **81.77** | **78.49** | **86.64** | **74.04** | 64.15 | 80.53 | **71.42** | 60.72 | **77.03** | 68.96 | **58.29** |
| mix(en, en-zh) | Original | 95.61 | 71.13 | 65.80 | 82.29 | 65.44 | 60.53 | 76.93 | 62.36 | 53.87 | 75.93 | 61.67 | 53.35 |
| | CS-baseline | 94.67 | 73.76 | 57.92 | 81.86 | 67.79 | 55.41 | 78.40 | 65.58 | 52.74 | 74.39 | 62.67 | 49.57 |
| | Shared $Q_{cross}$ | 96.56 | **77.10** | **74.84** | 84.52 | **71.96** | 61.35 | **81.39** | **71.31** | **63.25** | 75.61 | 67.42 | 55.40 |
| | Pair $Q_{cross}$ | **96.71** | 75.00 | 72.39 | **87.55** | 71.83 | **62.88** | 80.09 | 71.08 | 60.04 | **76.03** | **68.70** | **61.42** |

Table 3: Average cross-lingual transfer of mBERT and XLM-R models on **RuleTaker** datasets to monolingual samples (mono) and code-switched language pairs (en-X and X-en). The *original* is the pre-trained model and the *CS-baseline* is the model that pre-trained on code-switched data. Shared $Q_{cross}$ and Pair $Q_{cross}$, refer to cases where the cross-lingual query matrix $Q_{cross}$ is either shared across many language pairs or is specific to each language pair, respectively.

the X-en format. We also perform a qualitative analysis of self-attention patterns for our proposed method in Figure 5, showing that the attention patterns remain more similar between in-language and cross-lingual code-switched samples (unlike Figure 4). We hypothesize that the attention pattern stability makes the MultiLM more *language-neutral*.

Regarding the cross-lingual transfer across languages, we observe that the reasoning ability of the model is not transferred across languages equally (Appendix F). The more similar the languages, the higher the transfer performance is. For example, the model trained on en-fr has its highest transfer performance in Latin languages (*e.g.*, es, it, en-es, and en-it). For almost all cases, and regardless of the training data language, en-fa and en-ar are the hardest languages to transfer to.

To study the effect of the cross-lingual query matrix on an implicit reasoning task, we expand our experimentation to include the LeapOfThought (LoT)

dataset. Table 4 illustrates the average zero-shot transfer performance for this dataset. For this dataset, our proposed method also enhances the reasoning ability of the models for all examined language pairs. However, the degree of improvement observed is smaller compared to the Rule-Taker dataset. In the case of the implicit reasoning task within the LoT dataset, the model must rely on both contextual cues and implicit knowledge to successfully solve the task. Conversely, for the RuleTaker dataset, the model is required to fully reason over the context. Consequently, for implicit reasoning, the model only partially uses contextual information, resulting in a lesser impact on performance when improving cross-lingual context-question attentions.

## 4.4 Generalization to other Reasoning Tasks

So far, our experiments have focused on the logical reasoning ability of MultiLMs, either in monolin-

| Source Data | Method | mBERT | | | XLM-R | | |
|---|---|---|---|---|---|---|---|
| | | mono | en-X | X-en | mono | en-X | X-en |
| mix(en, en-fr) | Original | 65.71 | 71.89 | 67.69 | 69.81 | 73.39 | 71.70 |
| | CS-baseline | 62.18 | 66.92 | 62.79 | 70.00 | 70.92 | 69.48 |
| | Shared $Q_{cross}$ | **69.61** | 73.27 | 71.45 | 69.87 | **74.51** | 72.22 |
| | Pair $Q_{cross}$ | 67.95 | **75.79** | 71.13 | **71.12** | 74.20 | **73.09** |
| mix(en, en-de) | Original | 68.05 | 74.51 | 70.53 | 69.97 | 71.77 | 71.48 |
| | CS-baseline | 63.07 | 67.78 | 64.25 | 69.58 | 72.57 | 70.19 |
| | Shared $Q_{cross}$ | 67.48 | 75.52 | 71.52 | 70.00 | 73.22 | 72.07 |
| | Pair $Q_{cross}$ | **69.09** | **76.17** | **72.62** | 70.03 | **73.55** | **72.75** |
| mix(en, en-ru) | Original | 67.46 | 73.87 | 67.88 | 70.28 | 71.60 | 70.82 |
| | CS-baseline | 62.37 | 68.03 | 62.48 | 70.11 | 73.85 | 70.18 |
| | Shared $Q_{cross}$ | 67.84 | 74.59 | 69.85 | 70.10 | 73.20 | 71.91 |
| | Pair $Q_{cross}$ | **68.57** | **76.07** | **71.99** | **70.34** | **74.63** | **72.42** |
| mix(en, en-zh) | Original | 67.99 | 73.62 | 70.52 | 70.05 | 72.27 | **72.80** |
| | CS-baseline | 64.25 | 67.84 | 64.61 | 69.96 | 72.42 | 70.23 |
| | Shared $Q_{cross}$ | **69.19** | 74.88 | 71.45 | 70.20 | 73.00 | 72.15 |
| | Pair $Q_{cross}$ | 69.08 | **76.38** | **72.96** | 70.24 | **73.28** | 72.51 |

Table 4: Average cross-lingual transfer of mBERT and XLM-R on **LoT** dataset to monolingual samples (mono) and code-switched language pairs (en-X and X-en). The *original* is the pre-trained model and *CS-baseline* refers to the model pre-trained on code-switched data. Shared $Q_{cross}$ and Pair $Q_{cross}$, refer to cases where cross-lingual query is either shared across many language pairs or is specific to each language pair, respectively.

| Source Data | Model | mono | en-X | X-en |
|---|---|---|---|---|
| en | Original | **69.42** | 63.16 | 68.79 |
| mix(en, en-fr) | Original | 68.35 | 67.43 | 65.18 |
| | CS-baseline | 68.26 | 70.58 | 65.89 |
| | Shared $Q_{cross}$ | 68.30 | 69.22 | 70.16 |
| | Pair $Q_{cross}$ | 69.31 | **71.53** | **72.40** |

Table 5: Performance (accuracy) of mBERT model for the XNLI dataset in both monolingual and code-switched evaluation settings.

gual or code-switched settings. However, to demonstrate the proposed method's generalization to other reasoning tasks, we extend our experiments to the XNLI dataset. To create *structured code-switched* inputs for this task, we change the language of the *premise* and the *hypothesis* for a given input. More specifically, in a code-switched setting (*e.g.*, en-fr), the *premise* is in English, and the *hypothesis* is in French. We fine-tune the mBERT model on a combination of EN and code-switched EN and FR data (mix(en, en-fr)), then zero-shot transfer it to other languages for monolingual evaluations (excluding en and fr) and other language pairs for code-switched evaluation (excluding en-fr and fr-en pairs). Table 5 presents the performance of the mBERT model with the cross-lingual query compared to the baselines in both monolingual and code-switched settings. We observe ∼4% improvement on en-X, ∼7% on X-en, and competitive performance on monolingual evaluation setups, indicating the effectiveness of our proposed method on downstream tasks other than logical reasoning.

# 5 Related Work

**Reasoning in NLP.** Language models (LMs) have demonstrated their ability to perform logical reasoning over natural language statements (Clark et al., 2020b; Chen et al., 2023). They can also leverage their implicit knowledge for reasoning purposes (Talmor et al., 2020) and exhibit multi-step reasoning capabilities by utilizing chain-of-thought prompting, even with minimal demonstrations or instructions, when the model size is sufficiently large (Wei et al., 2022b; Kojima et al., 2022; Wei et al., 2022a; Tang et al., 2023). In parallel to English-centric efforts on reasoning tasks, there have been attempts to create multilingual reasoning datasets to evaluate the cross-lingual abilities of pre-trained MultiLMs (Conneau et al., 2018b; Artetxe et al., 2020; Clark et al., 2020a; Hu et al., 2020; Shi et al., 2022). Recent pre-trained large MultiLMs like BLOOM (Scao et al., 2022), BLOOMZ (Muennighoff et al., 2022), and XGLM (Lin et al., 2021), exhibited promising few-shot performance on a variety of cross-lingual reasoning datasets using in-context learning (Brown et al., 2020). Prior works studied the reasoning ability of MultiLMs in the context of open-retrieval answer generation (Asai et al., 2021), and mathematical problem-solving in a multilingual setting via chain-of-thought reasoning (Shi et al., 2022). This work conducts the first investigation of the logical reasoning capability of MultiLMs and proposes a cross-lingual-aware attention mechanism to improve their performance.

**Cross-lingual Transfer.** MultiLMs such as mBERT (Devlin et al., 2019), XLM-R (Conneau et al., 2020), mT5 (Xue et al., 2021), and XGLM (Lin et al., 2021) have achieved state of the art results in cross-lingual understanding tasks by jointly pre-training Transformer models (Vaswani et al., 2017) on many languages. These models have shown effective cross-lingual transfer for many tasks, including named entity recognition (Pires et al., 2019b; Wu and Dredze, 2019; Foroutan et al., 2022), cross-lingual natural language inference (Conneau et al., 2018a; Hu et al., 2020), question answering (Lewis et al., 2019), and commonsense reasoning (Tikhonov and Ryabinin, 2021). This study focuses on the cross-lingual transfer performance of MultiLMs in the context of logical reasoning.

**Code-switched NLP.** Code-switching is a linguistic phenomenon of alternating between two

or more languages within a single conversation or text. In recent years, code-switching-related research has been growing in the NLP community. The growth is motivated by the increasing need for NLP systems to handle code-switched data and call to pay more attention to multilingualism and low-resource languages (Doğruöz et al., 2021; Winata et al., 2022; Jose et al., 2020; Sitaram et al., 2019). Previous research has been done for a diverse range of tasks such as Language Identification, Part of Speech Tagging, Sentiment Analysis, and Automatic Speech Recognition (Winata et al., 2021; Khanuja et al., 2020; Ostapenko et al., 2022; Tarunesh et al., 2021). To the best of our knowledge, this work is the first to study logical reasoning in the context of code-switched NLP. Furthermore, a majority of prior studies have focused on word-level code-switching, where the language of certain words in a text randomly changes. However, our investigation delves into the realm of "structured code-switching", wherein language transitions occur at a section level.

## 6 Discussion

In this study, we explored the effectiveness of MultiLMs in a code-switched setting and found that while these models exhibit strong reasoning capabilities in monolingual settings, they struggle when it comes to code-switching. To address this, we first proposed the *structured* attention dropout, which encourages the model to rely less on cross-lingual attention when dealing with code-switched data. This simple method considerably improved cross-lingual transfer to other code-switched languages, demonstrating the importance of structured attention for this setting. We then proposed a novel *structured* attention mechanism, incorporating the *cross-lingual query*, that helps the model to better handle cross-lingual attention in the code-switched setting. The proposed cross-lingual query matrix, pre-trained on unsupervised code-switched data, significantly improved the cross-lingual transfer to other code-switched language pairs in all studies settings, demonstrating the importance of code-switched *alignment* for MultiLMs. We also observed better cross-lingual code-switched performance for the LeapOfThought dataset (real-world knowledge contexts) compared to RuleTaker (utilizing artificial facts and rules). We attribute LeapOfThought's better code-switched performance to the usage

of real-world knowledge in the reasoning context (compared to artificial facts and rules in RuleTaker), in line with Tang et al. (2023) observation that language models perform better when provided with commonsense-consistent context, and struggle with artificial ones.

## 7 Limitations

In this work, we evaluate our proposed method on encoder-only language models, and the impact of this method on autoregressive models and encode-decoder-only models has not been explored, leaving room for further investigation and evaluation. Moreover, our experiments are limited to relatively small language models (less than one billion parameters), and the results and our findings do not necessarily extend to large language models. Furthermore, we should highlight that the scope of our experiments is constrained by the availability of multilingual data and computational resources. Consequently, our evaluation is limited to two specific datasets and covers only nine languages. While we strive for diversity in our selection, it is important to recognize that broader and more extensive datasets encompassing a wider range of languages could offer additional perspectives and potentially reveal new insights.

## Acknowledgments

We thank Debjit Paul, Syrielle Montariol, Angelika Romanou, and Deniz Bayazit for their helpful discussions and feedback on our paper. We also gratefully acknowledge the support of the Swiss National Science Foundation (No. 215390), Innosuisse (PFFS-21-29), the EPFL Science Seed Fund, the EPFL Center for Imaging, Sony Group Corporation, and the Allen Institute for AI.

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

# A   Dataset Details

This section further elaborates on the datasets that are used in our experiments. Both datasets in this study are translated using Google Translate API to investigate our proposed method's cross-lingual transfer. Starting from the English dataset, the samples are translated into eight other languages, namely, French (Fr), Farsi (Fa), German (De), Arabic (Ar), Spanish (Es), Chinese (Zh), Russian (Ru), and Italian (It). Below we discuss in more detail each studied dataset.

## A.1   RuleTaker Dataset

RuleTaker dataset (Clark et al., 2020b) is a set of five datasets, requiring various depths of inference to answer the questions. Each dataset consists of examples in the form of a triple: (context, statement, answer). The context is composed of a series of facts and rules, while the statement represents the question that needs to be proven and the answer is either "T" (true) if the statement logically follows from the context, or "F" (false) if it does not (false under a closed-world assumption, CWA). All the facts, rules, and question statements are expressed in synthetic English. Essentially, each example represents a self-contained logical theory in linguistic form, with a question asking, "Is it true?" The dataset generation procedure ensures that every question can be answered by a formal reasoner, given the closed-world assumption (CWA).

Each dataset limited by the maximum level of inference needed to validate the facts employed in its corresponding questions. These datasets are categorized based on their depth restrictions (up to depths $D = 0$, $D \leq 1$, $D \leq 2$, $D \leq 3$ and $D \leq 5$ respectively). A depth of $D = 0$ implies that the accurate facts can be readily "proven" by directly looking them up within the given context, without requiring any inference. The fifth dataset, encompasses questions that span up to a depth of 5. This dataset serves as a test to assess the ability to generalize to depths not encountered during training on the other four datasets. In our experiments, we use datasets with depths 0 to 4. Each dataset contains 100k examples randomly split 70/10/20 into train/dev/test partitions.

## A.2   LeapOfThought (LoT) Dataset

The primary focus of the LoT dataset (Talmor et al., 2020) revolves around a specific form of inference that integrates implicit taxonomic knowledge (such

| Experiment | EXPLICIT REASONING | IMPLICIT REASONING |
|---|---|---|
| Context-only | 94.56 | 68.98 |
| Subject swap | 83.08 | 51.46 |
| Object swap | 91.85 | 53.6 |
| Subject & Object swap | 87.5 | 52.36 |

Table 6: This table investigates the artifacts present in the LeapOfThought dataset. We evaluate the model's performance when either different parts of a given sample are removed (*e.g.*, context-only model) or different noises are injected into the statement (*e.g.*, swapping the subject in the statement with a randomly selected entity). The experiments that involve swapping entities in the statement are performed on the modified version of LoT, as discussed in section A.2.1.

as hypernymy and meronymy) with explicit rules derived from natural language. The hypernymy relations and properties are derived from WORD-NET (Fellbaum, 1998) and CONCEPTNET (Speer et al., 2017). Each example consists of two components: (1) a hypothesis, which is a textual statement that can be either true or false, and (2) explicit knowledge, represented as a list of textual statements. These statements can be classified as either facts, which describe properties of specific entities, or rules, which describe properties of a particular class. The explicit knowledge is carefully constructed to ensure that the truth value of the hypothesis cannot be determined solely based on the provided information. It necessitates the inclusion of additional knowledge encoded within the language model. This dataset contains two main test sets; EXPLICIT REASONING which performs inference over explicit natural language statements, and IMPLICIT REASONING where the model must reason by combining the context with missing information that should be implicitly encoded by the model. The dataset consists of 30,906 training examples, 1,289 development examples, and 1,289 test examples.

### A.2.1 Discussion on LoT Dataset Artifacts

LoT dataset was designed to test how well NLP models can (possibly) reason using real-world knowledge. However, as we show in this section, the dataset has some artifacts that causes the NLP models to take shortcuts instead of actually performing the reasoning. In the following analysis, we only focus on the original English LoT dataset (and not on the translated samples).

In our preliminary experiments on this dataset, we observed that MultiLMs perform surprisingly high in cross-lingual code-switched settings (on

EXPLICIT dev set), even if the statement is in a medium-resource language like Farsi or Arabic (context being in English). We hypothesized that the model is mostly relying on the context for reasoning; therefore, the statement being in a medium/low-resource language does not necessarily impact the model's performance. We validate this hypothesis by training a context-only model (without having access to respective statements), and surprisingly this model performs ∼94% on the EXPLICIT dev set (see Table 6). In order to ensure that the model can not get non-random accuracy by relying only on the context, we randomly negate 50% of statements (also negating the respective labels), so that a context-only model would perform randomly. The resulting dataset is the *modified* LoT that is used in all experiments in the paper.

In order to further investigate artifacts present in the modified LoT dataset, we inject noise into the statement (without changing the context) as following:

- Swapping statement's subject with a randomly selected entity from the whole dataset

- Swapping statement's object with a randomly selected entity from the whole dataset

- Swapping statement's subject and object with a randomly selected entity from the whole dataset

As demonstrated in Table 6, given the EXPLICIT evaluation results, the model can still get high reasoning performance even when the entities in the context and statement are not consistent. However, as reasoning performance on IMPLICIT evaluation set drops to (almost) random when noise are injected into the statement entities, we believe that LoT artifacts have less effect on this evaluation setting. Therefore, to evaluate the MultiLM's reasoning performance, we use the IMPLICIT evaluation set throughout the paper.

## B Multilingual Reasoning: XLM-R results

Sections 3.2 and 3.3 discussed the in-language and cross-lingual performance of the mBERT model on monolingual and code-switched data. This section evaluates the XLM-R model on the same evaluation settings as mBERT.

Table 7 demonstrates the average in-language and cross-lingual zero-shot performance of XLM-R

| | RuleTaker | | | | | | | | LeapOfThought | |
| | Depth-0 | | Depth-1 | | Depth-2 | | Depth-3 | | | |
| | in-lang. | cross-ling. | in-lang. | cross-ling. | in-lang. | cross-ling. | in-lang. | cross-ling. | in-lang. | cross-ling. |
|---|---|---|---|---|---|---|---|---|---|---|
| en | 100.00 | 94.95 | 87.23 | 77.75 | 87.24 | 76.26 | 82.78 | 71.21 | 76.70 | 70.36 |
| fr | 99.40 | 95.83 | 87.10 | 81.53 | 83.92 | 77.78 | 81.33 | 73.78 | 74.64 | 66.68 |
| fa | 100.00 | 90.05 | 87.65 | 80.49 | 85.46 | 72.97 | 80.70 | 67.41 | 72.11 | 69.07 |
| de | 99.33 | 94.73 | 85.48 | 80.10 | 84.92 | 79.08 | 81.15 | 73.60 | 78.32 | 69.09 |
| ar | 99.13 | 85.95 | 84.65 | 76.29 | 84.43 | 71.13 | 81.13 | 66.68 | 69.70 | 71.71 |
| es | 99.96 | 94.46 | 90.53 | 82.00 | 84.34 | 75.61 | 83.20 | 71.83 | 80.60 | 71.25 |
| zh | 99.96 | 86.03 | 85.99 | 77.63 | 82.56 | 69.10 | 81.95 | 67.69 | 82.62 | 66.35 |
| ru | 99.81 | 93.36 | 87.20 | 77.37 | 82.34 | 70.16 | 81.35 | 73.40 | 73.47 | 71.36 |
| it | 99.75 | 92.18 | 87.21 | 78.38 | 84.72 | 78.41 | 82.85 | 74.34 | 73.89 | 72.29 |
| Average | 99.70 | 91.95 | 87.00 | 79.06 | 84.44 | 74.50 | 81.83 | 71.10 | 75.78 | 69.80 |

Table 7: **Monolingual Setting:** In-language and cross-lingual zero-shot performance (accuracy) of the XLM-R model for the RuleTaker and LeapOfThought datasets.

| | RuleTaker | | | | | | | | | | | | LeapOfThought | | |
| | Depth-0 | | | Depth-1 | | | Depth-2 | | | Depth-3 | | | | | |
| | in-lang. | en-X | X-en | in-lang. | en-X | X-en | in-lang. | en-X | X-en | in-lang. | en-X | X-en | in-lang. | en-X | X-en |
|---|---|---|---|---|---|---|---|---|---|---|---|---|---|---|---|
| en-fr | 98.47 | 54.18 | 52.38 | 88.18 | 61.10 | 57.61 | 85.54 | 54.54 | 50.45 | 83.14 | 52.50 | 50.66 | 84.02 | 78.41 | 75.40 |
| en-fa | 98.61 | 63.70 | 55.51 | 88.99 | 60.63 | 56.39 | 85.90 | 60.59 | 56.32 | 74.38 | 54.92 | 52.78 | 85.96 | 82.57 | 72.06 |
| en-de | 99.62 | 59.59 | 52.98 | 92.68 | 58.41 | 53.06 | 85.17 | 57.25 | 55.14 | 86.09 | 54.75 | 50.45 | 87.20 | 80.55 | 74.45 |
| en-ar | 98.99 | 60.36 | 57.55 | 76.16 | 57.40 | 50.58 | 83.65 | 62.21 | 55.45 | 70.12 | 56.21 | 51.20 | 80.84 | 77.06 | 70.16 |
| en-es | 100.00 | 60.01 | 52.32 | 92.59 | 63.06 | 56.99 | 87.97 | 59.17 | 53.61 | 77.48 | 55.70 | 51.33 | 86.89 | 81.30 | 77.60 |
| en-zh | 99.98 | 62.36 | 54.45 | 87.00 | 63.01 | 58.23 | 85.55 | 58.78 | 56.99 | 82.96 | 57.59 | 53.71 | 88.75 | 82.13 | 79.24 |
| en-ru | 99.93 | 64.07 | 55.85 | 80.11 | 64.59 | 51.57 | 86.85 | 57.40 | 51.32 | 85.35 | 56.95 | 48.85 | 81.85 | 78.55 | 74.13 |
| en-it | 99.89 | 64.16 | 56.25 | 89.11 | 61.03 | 54.45 | 85.94 | 60.31 | 53.39 | 85.08 | 56.80 | 50.73 | 80.44 | 74.96 | 70.41 |
| Average | 99.44 | 61.05 | 54.66 | 86.85 | 61.15 | 54.86 | 85.82 | 58.78 | 54.08 | 80.58 | 55.68 | 51.21 | 84.49 | 79.44 | 74.18 |

Table 8: **Code-Switched Setting:** In-language and cross-lingual performance (accuracy) of the XLM-R model for the RuleTaker and LeapOfThought datasets.

| | | RuleTaker | | | | | |
| | | Depth-0 | | | Depth-1 | | |
| Training Language | Training Method | mono | en-X | X-en | mono | en-X | X-en |
|---|---|---|---|---|---|---|---|
| mix(en, en-fr) | Full FT | 87.57 | 63.05 | 59.40 | 68.95 | 59.72 | 58.03 |
| | Bitfit | 89.14 | 65.38 | 60.81 | 70.76 | 60.48 | 58.16 |
| mix(en, en-zh) | Full FT | 91.00 | 62.85 | 56.56 | 75.30 | 62.94 | 57.02 |
| | Bitfit | 91.20 | 65.58 | 59.80 | 76.43 | 63.02 | 57.20 |
| mix(en, en-de) | Full FT | 89.72 | 62.00 | 58.42 | 79.13 | 58.34 | 56.67 |
| | Bitfit | 88.71 | 66.75 | 59.10 | 68.98 | 58.64 | 56.69 |
| mix(en, en-ru) | Full FT | 90.22 | 63.17 | 56.73 | 73.33 | 59.94 | 55.65 |
| | Bitfit | 91.69 | 69.25 | 60.23 | 76.49 | 60.40 | 57.09 |

Table 9: Performance of fully fine-tuned versus Bitfit-tuned mBERT models on the RuleTaker dataset. Bitfit-tuned models perform better or competitively to the fully fine-tuned setting, especially on the code-switched evaluation setups.

for each source language in a monolingual setting. Code-switched evaluation results are depicted in Table 8.

# C Experimental Setup Details

## C.1 Full Fine-tuning Versus Bitfit

As discussed in section 4.2, our proposed model and the baselines' performances in Tables 4 and 3 are achieved by Bitfit tuning (Zaken et al., 2021). It has been previously observed by Tu et al. (2022) that parameter-efficient fine-tuning (PEFT) has

better cross-lingual generalization than full fine-tuning. In our experiments, we also found out that using a PEFT method like Bitfit considerably improves our cross-lingual transfer across different languages.

Table 9 demonstrates the generalization improvement brought by Bitfit over full fine-tune baseline for the RuleTaker dataset, especially in code-switched settings. We observed similar pattern for other RuleTaker depths and the LoT dataset. It is worth noting that using a PEFT method especially helps with transfer to code-switched tasks, which is our main focus in this paper.

## C.2 Curriculum Learning

For depths 2 and 3 of RuleTaker dataset, which involves more reasoning hops, we observed that curriculum learning (Bengio et al., 2009) makes the XLM-R training more robust. The curriculum learning is performed by first training the MultiLM for 3 epochs on a subset of dataset that has depth 0 (*i.e.*, no hop is needed for reasoning), and then the training is continued on the full dataset. This technique not only makes the XLM-R training more robust, but also improves the final reasoning per-

formance.

## C.3 Hyperparameters

The hyperparameters for all the experiments is provided in Table 10 for both mBERT and XLM-R models. We use the AdamW optimizer with a warmup ratio of 0.1 for all experiments.

## D  Cross-lingual Query

This section further discusses the methods proposed in section 4.1.

### D.1  Structured Attention Dropout

As previously discussed in section 4.1, limiting the cross-lingual attention in the fine-tuning makes this phase more consistent with the pre-training, where the MultiLM mostly deals with monolingual attentions. Table 11 demonstrates that applying dropout on cross-lingual attenions (see Figure 2) considerably improves cross-lingual generalization in code-switched settings. Table 11 results are achieved by a 40% dropout on cross-lingual attentions (*i.e.*, $P_{mask} = 0.4$)

### D.2  Interfering Cross-lingual Query

Inspired by the promising performance of the structured attention dropout, we propose a setting where the query matrix Q also partially handles the cross-lingual attentions, and cross-lingual query $Q_{cross}$ partially handles monolingual attentions. The only difference between the interfering cross-lingual query and the non-interfering scheme is their respective attention masks, $M_1$ and $M_2$, as illustrated in Figure 3. We also empirically demonstrate in Table 12 that the interfering scheme consistently performs better generalization than the non-interfering one, especially in the code-switched settings. For all the fine-tuning experiments with the interfering cross-lingual query, we use a 70% attention dropout (*i.e.*, $P_{mask} = 0.7$), meaning that 70% of cross-lingual attentions for query Q, and 70% of monolingual attentions for query $Q_{cross}$ are masked.

## E  Attention Visualization

As discussed earlier, MultiLMs perform well on in-language, but when they are transferred to other languages (especially code-switched languages) their performance hinders considerably (see Table 2). This section first analyzes the attention pattern of baseline models, both on in-language and cross-lingual evaluation settings. Then, we analyze the

attention pattern of our proposed model which incorporates cross-lingual query.

We hypothesize that in order to have a reasonable cross-lingual performance, the cross-lingual samples' attention pattern should not change significantly compared to the in-language samples. Figure 4 visualizes the attention pattern between tokens in the last (baseline) mBERT layer across all attention heads. The mBERT model is fine-tuned on the mix(en, en-fr) depth-0 of RuleTaker dataset, so the en-fr sample is considered in-language and the en-ar sample is considered a zero-shot transfer. It is worth noting the two samples are semantically the same and only the questions are in different languages. Comparing the two samples' attention patterns, we can see that the attention pattern considerably changes (especially the strong attention signals getting much weaker when en-ar sample is given as input), which to some extent explains the poor generalization of the baseline models to other code-switched tasks.

In contrast, as demonstrated by Figure 5, the attention pattern of our proposed method, which incorporates cross-lingual query, is much more stable between in-language (*i.e.*, en-fr sample), and the zero-shot transfer (*i.e.*, en-ar sample). We believe that the observed stability in the attention patterns makes our models more *language-neutral* compared to the baseline, which is also demonstrated by the significant cross-lingual improvements over the baselines in Tables 3 and 4.

## F  Detailed Cross-lingual Query Results

Tables 3 and 4 demonstrated the average cross-lingual transfer to either monolingual or code-switched settings. This section demonstrates the detailed cross-lingual performance of models with cross-lingual query and the *Original* and *CS-baseline*. Tables 13 and 14 present the detailed cross-lingual transfer of mBERT trained on the RuleTaker and LeapOfThought datasets, respectively. Tables 15 and 16 present similar detailed cross-lingual performance of XLM-R model on the RuleTaker and LeapofThought datasets, respectively.

| mBERT | | | | | | |
|---|---|---|---|---|---|---|
| Train Method | Dataset | Epoch / Iteration | Batch Size | Learning Rate | Evaluation Metric | Attention Dropout Probability |
| Full FT | RuleTaker | 5 | 32 | 1e-5 | Accuracy | N/A |
| | LeapOfThought | 10 | 32 | 1e-5 | Accuracy | N/A |
| BitFit | RuleTaker | 35 | 32 | 4e-4 | Accuracy | 0.7 |
| | LeapOfThought | 30 | 32 | 4e-4 | Accuracy | 0.7 |
| Pre-training Q | XNLI dataset | 500,000 | 16 | 2e-5 | Perplexity | 1.0 |

| XLM-R | | | | | | |
|---|---|---|---|---|---|---|
| Train Method | Dataset | Epoch / Iteration | Batch Size | Learning Rate | Evaluation Metric | Attention Dropout Probability |
| Full FT | RuleTaker | 5 | 32 | 5e-6 | Accuracy | N/A |
| | LeapOfThought | 10 | 32 | 5e-6 | Accuracy | N/A |
| BitFit | RuleTaker | 35 | 32 | 3e-4 | Accuracy | 0.7 |
| | LeapOfThought | 30 | 32 | 4e-4 | Accuracy | 0.7 |
| Pre-training Q | XNLI dataset | 500,000 | 8 | 2e-6 | Perplexity | 1.0 |

Table 10: Hyperparameters of the pre-training and fine-tuning experiments for mBERT and XLM-RoBERTa models. Learning rate decays linearly from the initial value to zero.

| Training Method | Drop attention | Transfer Setting | | |
|---|---|---|---|---|
| | | Monolingual | en-X | X-en |
| Full Fine-tune | Yes | **90.00** | **65.57** | **62.74** |
| | No | 89.48 | 62.68 | 59.53 |
| Bitfit | Yes | 82.98 | **73.20** | **68.24** |
| | No | **89.14** | 65.38 | 60.81 |

Table 11: Average cross-lingual transfer of mBERT model when tuned on a mixture of English and English-French (mix(en, en-fr)) RuleTaker dataset (depth-0). The (zero-shot) cross-lingual transfer to code-switched tasks gets considerably better with *structured* attention dropout (see section 4.1), either in full fine-tune or Bitfit (Zaken et al., 2021) tuning.

| Training Method | Interfering | Transfer Setting | | |
|---|---|---|---|---|
| | | Monolingual | en-X | X-en |
| Bitfit | Yes | 93.65 | 77.79 | 68.27 |
| | No | 91.96 | 73.08 | 66.28 |

Table 12: Average cross-lingual transfer of mBERT model when fine-tuned on a mixture of English and English-French (mix(en, en-fr)) RuleTaker dataset (depth-0). Both models incorporate language-pair-specific cross-lingual query (*i.e.*, Pair $Q_{cross}$) and are trained with Bitfit tuning. The only difference between the two runs is whether an interfered version of the cross-lingual query is used or not. We can observe that the interfered variant consistently outperforms the other variant, in monolingual and code-switched settings.

| Train Data | Method | en | fr | fa | de | ar | es | zh | ru | it | en-fr | en-it | en-es | en-zh | en-ru | en-de | en-fa | en-ar | fr-en | de-en | fa-en | es-en | it-en | ru-en | zh-en | ar-en |
|---|---|---|---|---|---|---|---|---|---|---|---|---|---|---|---|---|---|---|---|---|---|---|---|---|---|---|
| **RuleTaker Depth-0** | | | | | | | | | | | | | | | | | | | | | | | | | | |
| mix(en, en-fr) | Original | 99.97 | 94.76 | 70.59 | 93.28 | 87.21 | 93.84 | 80.48 | 89.21 | 92.89 | 97.23 | 77.52 | 72.93 | 50.31 | 55.36 | 69.25 | 49.27 | 51.13 | 80.07 | 65.03 | 53.01 | 63.72 | 65.52 | 53.48 | 53.14 | 52.47 |
| | CS-baseline | 99.95 | 83.24 | 55.16 | 78.65 | 74.05 | 84.97 | 69.79 | 81.29 | 83.30 | 99.25 | 72.75 | 84.19 | 79.79 | 72.36 | 76.49 | 48.41 | 64.51 | 59.69 | 55.53 | 51.90 | 58.16 | 57.86 | 53 | 51.53 | 52.15 |
| | Shared $Q_{cross}$ | 100 | 96.29 | 77.19 | 94.48 | 95.09 | 95.96 | 85.81 | 92.71 | 95.18 | 98.73 | 83.33 | 76.01 | 62.39 | 60.62 | 74.96 | 50.33 | 54.16 | 93.32 | 70.52 | 52.71 | 69.38 | 73.89 | 55.04 | 53.94 | 52.50 |
| | Pair $Q_{cross}$ | 99.99 | 97.04 | 81.05 | 94.97 | 94.92 | 96.69 | 89.05 | 92.87 | 96.23 | 99.07 | 83.44 | 86.13 | 81.04 | 72.65 | 80.61 | 52.19 | 67.15 | 92.84 | 72.04 | 55.78 | 71.74 | 75.86 | 61.97 | 59.58 | 56.36 |
| mix(en, en-de) | Original | 99.94 | 94.41 | 73.12 | 93.18 | 85.04 | 92.51 | 79.71 | 86.44 | 94.01 | 76.31 | 67.79 | 71.61 | 56.14 | 61.78 | 95.92 | 51.22 | 53.26 | 68.52 | 73.02 | 52.25 | 58.65 | 61.83 | 52.97 | 53.82 | 51.75 |
| | CS-baseline | 99.97 | 87.99 | 67.21 | 86.12 | 81.59 | 88.73 | 80.73 | 82.72 | 87.84 | 84.58 | 69.56 | 80.47 | 78.06 | 72.66 | 99.35 | 50.54 | 62.59 | 65.21 | 64.23 | 52.63 | 59.64 | 57.61 | 52.71 | 51.95 | 52.49 |
| | Shared $Q_{cross}$ | 99.98 | 94.42 | 82.77 | 94.34 | 89.09 | 94.26 | 82.84 | 91.24 | 93.59 | 80.38 | 73.48 | 75.31 | 63.82 | 64.35 | 99.33 | 51.13 | 52.97 | 74.99 | 90.72 | 54.95 | 65.14 | 70.36 | 54.86 | 54.40 | 52.83 |
| | Pair $Q_{cross}$ | 99.99 | 94.77 | 91.58 | 95.51 | 94.09 | 95.58 | 86.69 | 93.21 | 95.60 | 85 | 72.30 | 82.55 | 82.16 | 71.11 | 99.18 | 53.97 | 64.31 | 79.12 | 93.83 | 56.70 | 72.52 | 71.60 | 61.77 | 65.96 | 57.32 |
| mix(en, en-ru) | Original | 99.81 | 94.96 | 83.31 | 92.29 | 86.06 | 94.88 | 90.80 | 89.38 | 93.70 | 71.81 | 70.46 | 68.73 | 70.54 | 92.55 | 65.02 | 60.04 | 54.86 | 68.33 | 64.27 | 54.58 | 62.68 | 65.98 | 58.70 | 54.89 | 52.44 |
| | CS-baseline | 99.97 | 88.43 | 64.30 | 81.49 | 75.32 | 89.72 | 82.50 | 78.64 | 92.52 | 81.63 | 65.88 | 79.35 | 80.30 | 99.82 | 80.49 | 54.16 | 65.71 | 62.71 | 56.56 | 50.59 | 59.87 | 57.14 | 51.01 | 49.74 | 49.79 |
| | Shared Q | 99.98 | 95.02 | 84.43 | 93.75 | 91.28 | 95.63 | 90.07 | 93.94 | 94.92 | 77.52 | 77.11 | 76.42 | 79.19 | 98.22 | 77.40 | 63.67 | 60.24 | 75 | 70.23 | 63.74 | 73.14 | 75.09 | 79.92 | 66.65 | 59 |
| | Pair $Q_{cross}$ | 99.93 | 94.49 | 81.83 | 92.80 | 89.75 | 95.04 | 89.20 | 92.52 | 94.50 | 82.52 | 74.97 | 79.26 | 81 | 96.20 | 79.86 | 60.45 | 63.52 | 77.11 | 75.92 | 62.76 | 75 | 74.86 | 77.89 | 71.64 | 59.81 |
| mix(en, en-zh) | Original | 99.84 | 93.78 | 82.44 | 93.07 | 88.84 | 93.24 | 85.55 | 90 | 94.05 | 65.30 | 65.75 | 65.66 | 91.94 | 60.49 | 64.69 | 56.49 | 54.32 | 69.67 | 62.03 | 54.55 | 60.94 | 63.94 | 53.79 | 61.25 | 52.19 |
| | CS-baseline | 99.96 | 86.26 | 67.56 | 79.58 | 76.14 | 87.50 | 74.97 | 84.45 | 92.02 | 77.62 | 64.96 | 75.41 | 99.87 | 68.72 | 68.46 | 49.77 | 56.92 | 65 | 59.97 | 52.62 | 60.40 | 57.86 | 53.52 | 57.89 | 52.43 |
| | Shared $Q_{cross}$ | 99.99 | 94.50 | 88.33 | 92.97 | 94 | 95.45 | 91.37 | 91.62 | 95.08 | 74.57 | 68.44 | 67.16 | 98.50 | 63.80 | 66.63 | 54.49 | 54.34 | 74.88 | 65.03 | 57.79 | 67.57 | 67.85 | 57.84 | 72.83 | 52.96 |
| | Pair $Q_{cross}$ | 100 | 96.20 | 69.73 | 94.47 | 95.04 | 96.76 | 96.54 | 93.85 | 96.26 | 78.80 | 71.38 | 73.73 | 99.46 | 63.36 | 72.05 | 61.37 | 56.55 | 76.77 | 72.03 | 58.88 | 72.69 | 75.81 | 60.18 | 88.73 | 53.57 |
| **RuleTaker Depth-1** | | | | | | | | | | | | | | | | | | | | | | | | | | |
| mix(en, en-fr) | Original | 83.29 | 73.71 | 59.34 | 75.21 | 65.18 | 72.93 | 67.43 | 67.66 | 72.08 | 74.75 | 67.69 | 64.52 | 53.36 | 56.65 | 61.49 | 52.26 | 53.09 | 67.48 | 60.50 | 53.80 | 60.47 | 60.70 | 54.65 | 53.72 | 53.90 |
| | CS-baseline | 83.65 | 66.62 | 55.62 | 66.96 | 66.70 | 68.61 | 66.96 | 69.22 | 64 | 82.24 | 70.33 | 72.68 | 70.97 | 67.47 | 70.55 | 51 | 60.76 | 54.49 | 53.34 | 53.84 | 54.42 | 54.79 | 53.32 | 53.32 | 53.67 |
| | Shared $Q_{cross}$ | 87.18 | 81.53 | 70.64 | 82.66 | 69.82 | 78.72 | 73.42 | 76.41 | 79.10 | 81.93 | 75.49 | 72.80 | 64.30 | 61.12 | 71.04 | 55.97 | 55.46 | 78.42 | 67.77 | 56.12 | 67.13 | 70.16 | 58.19 | 58.65 | 54.99 |
| | Pair $Q_{cross}$ | 86.40 | 79.87 | 67.05 | 80.94 | 74.96 | 79.22 | 74.49 | 75.66 | 78.40 | 82.55 | 74 | 71.50 | 67.87 | 65.60 | 72.01 | 54.81 | 60.05 | 77.49 | 67.07 | 54.47 | 68.82 | 64.99 | 58.43 | 59.51 | 59.32 |
| mix(en, en-de) | Original | 77.91 | 71.26 | 61.15 | 65.61 | 67.52 | 69.42 | 68.95 | 68.58 | 70.42 | 62.88 | 62.07 | 60.85 | 52.77 | 57 | 66.66 | 53.17 | 53.71 | 61.22 | 61.10 | 53.71 | 58.83 | 57.40 | 53.98 | 53.58 | 53.67 |
| | CS-baseline | 82.98 | 68.85 | 59.59 | 64.13 | 66.65 | 68.75 | 64.24 | 68.58 | 68.97 | 74.74 | 64.09 | 71.19 | 72.36 | 68.07 | 80.06 | 51.73 | 60.76 | 56.01 | 55.92 | 52.59 | 54.64 | 53.88 | 53.04 | 53.29 | 52.57 |
| | Shared $Q_{cross}$ | 86.89 | 79.15 | 70.21 | 80.56 | 72.57 | 76.37 | 71.50 | 75.53 | 77.88 | 73.07 | 68.57 | 68.15 | 62.47 | 62.19 | 81.17 | 54.50 | 56.95 | 69.22 | 75.70 | 54.61 | 63.17 | 65.29 | 55.32 | 55.07 | 54.72 |
| | Pair $Q_{cross}$ | 87.91 | 80.22 | 71.21 | 81.78 | 64.43 | 80.57 | 74.70 | 75.59 | 80.04 | 70.59 | 67.53 | 70.22 | 69.72 | 66.45 | 83.90 | 54.81 | 65.57 | 70.17 | 74.52 | 55.41 | 65.67 | 65.76 | 58.81 | 59.66 | 55.77 |
| mix(en, en-ru) | Original | 85.87 | 79.91 | 67.78 | 78.80 | 71.32 | 78.69 | 74.22 | 73.22 | 78.63 | 63.94 | 62.49 | 60.76 | 56.85 | 68.16 | 61.17 | 55.21 | 54.60 | 62.68 | 60.03 | 53.87 | 58.57 | 59.33 | 54.82 | 53.69 | 53.72 |
| | CS-baseline | 84.22 | 74.01 | 54.92 | 70.21 | 69.50 | 73.90 | 68.54 | 70.34 | 73.86 | 76.23 | 66.88 | 75.69 | 71.53 | 81.13 | 71.96 | 51.02 | 65.26 | 57.70 | 57.21 | 53.55 | 57.50 | 56.07 | 54.72 | 53.54 | 53.60 |
| | Shared $Q_{cross}$ | 88.04 | 80.84 | 74.44 | 81.84 | 76.01 | 80.86 | 76.94 | 79.29 | 79.91 | 72.86 | 69.48 | 67.57 | 70.51 | 78.24 | 71.33 | 61.66 | 58.54 | 70.17 | 67.94 | 60.32 | 66.67 | 69.08 | 67.59 | 64.25 | 57.50 |
| | Pair $Q_{cross}$ | 87.16 | 80.48 | 71.71 | 81.04 | 71.75 | 79.72 | 76.69 | 76.69 | 80.24 | 75.32 | 73.22 | 73.22 | 75.39 | 98.27 | 78.42 | 58.68 | 59.25 | 72.08 | 66.35 | 63.99 | 67.77 | 71.51 | 62.33 | 56.32 | |
| mix(en, en-zh) | Original | 85.68 | 79.22 | 70.71 | 78.48 | 72.63 | 78.59 | 72.31 | 73.57 | 76.69 | 66.85 | 64.10 | 62.59 | 78.90 | 58.96 | 61.85 | 55.75 | 55.19 | 62.84 | 59.08 | 54.09 | 58.41 | 58.98 | 54 | 56.51 | 53.67 |
| | CS-baseline | 83.97 | 68 | 58.29 | 63.72 | 62.29 | 67.97 | 62.67 | 68.22 | 70.89 | 71.97 | 61.76 | 70.82 | 82.94 | 66.10 | 69.17 | 51.61 | 60.58 | 57 | 54.91 | 53.47 | 55.78 | 55.54 | 53.17 | 52.53 | 51.89 |
| | Shared $Q_{cross}$ | 86.21 | 79.80 | 59.52 | 77.54 | 63.58 | 79.94 | 75.66 | 75.62 | 78.09 | 69.18 | 67.32 | 65.25 | 78.71 | 63.18 | 65.31 | 56.47 | 55.43 | 69.61 | 63.24 | 56.59 | 63.98 | 58.52 | 64.84 | 55.42 | |
| | Pair $Q_{cross}$ | 88.28 | 80.15 | 66.84 | 81.34 | 74.99 | 80.83 | 79.34 | 77.24 | 81.40 | 70.56 | 61.40 | 68.84 | 80.82 | 69.33 | 70.10 | 55.42 | 60.51 | 67.52 | 68.17 | 54.20 | 70.38 | 60.43 | 60.98 | 73.86 | 58.25 |
| **RuleTaker Depth-2** | | | | | | | | | | | | | | | | | | | | | | | | | | |
| mix(en, en-fr) | Original | 84.82 | 71.80 | 54.28 | 73.04 | 56.09 | 70.97 | 62.55 | 62.30 | 71.06 | 79.10 | 66.76 | 64.55 | 56.26 | 58.84 | 60.94 | 54.90 | 55.74 | 67.77 | 57.66 | 50.75 | 57.30 | 57.77 | 51.33 | 50.89 | 50.94 |
| | CS-baseline | 83.53 | 62.26 | 57.33 | 58.71 | 60.59 | 62.25 | 60.03 | 61.07 | 62.67 | 81.92 | 68.33 | 74.34 | 66.16 | 65.90 | 70.33 | 51.48 | 61.52 | 54.76 | 52.97 | 50.81 | 54.43 | 54.82 | 50.86 | 51.98 | 50.14 |
| | Shared $Q_{cross}$ | 85.98 | 73.74 | 61.59 | 80.03 | 67.17 | 77.35 | 73.56 | 73.06 | 77.18 | 83.92 | 70.92 | 69.23 | 59.31 | 57.80 | 67.84 | 51.51 | 53.27 | 75.97 | 60.77 | 51.62 | 63.53 | 53.37 | 53.67 | 51.62 | |
| | Pair $Q_{cross}$ | 84.04 | 75.41 | 65.94 | 76.57 | 69.78 | 76.40 | 69.11 | 71.69 | 75.07 | 83.47 | 70.16 | 72.78 | 67.17 | 68.11 | 71.85 | 53.27 | 59.05 | 73.27 | 65.73 | 51.52 | 66.35 | 62.25 | 57.43 | 56.60 | 57.04 |
| mix(en, en-de) | Original | 84.26 | 75.64 | 65.94 | 76.86 | 68.72 | 74.06 | 72.14 | 70 | 72.89 | 68.23 | 64 | 65.52 | 56.05 | 62.60 | 78.85 | 54.34 | 53.43 | 62.07 | 62.35 | 51.37 | 57.40 | 58.09 | 51.50 | 51.64 | 50.84 |
| | CS-baseline | 83.87 | 63.27 | 57.76 | 59.33 | 59.59 | 62.55 | 60.24 | 59.44 | 66.15 | 74.53 | 63.31 | 73.54 | 69.79 | 66.76 | 81.65 | 50.69 | 59.47 | 52.53 | 50.35 | 50.13 | 54.17 | 52.53 | 50.63 | 50.74 | 50.26 |
| | Shared Q | 84.93 | 76.55 | 63.23 | 76.68 | 65.74 | 71.69 | 64.09 | 70.28 | 73.18 | 69.75 | 66.44 | 66.28 | 62.16 | 61.70 | 78.41 | 54.27 | 55.99 | 66.22 | 75.96 | 51.66 | 61.95 | 63.04 | 54.24 | 53.83 | 52.43 |
| | Pair $Q_{cross}$ | 85.28 | 76.46 | 65.63 | 77.02 | 65.65 | 74.10 | 75.35 | 71.04 | 73.56 | 73.23 | 67.34 | 70.09 | 69.34 | 68.51 | 82.13 | 56.34 | 60.39 | 67.45 | 75.18 | 53.65 | 62.06 | 63.92 | 59.92 | 58.64 | 56.48 |
| mix(en, en-ru) | Original | 80.34 | 71.59 | 60.20 | 72.79 | 63.99 | 69.44 | 66.38 | 66.99 | 69.34 | 55.82 | 56.72 | 56.25 | 55.90 | 54.33 | 54.03 | 55.96 | 54.33 | 54.03 | 52.36 | 50.79 | 53.16 | 53.11 | 51.43 | 51.79 | 50.18 |
| | CS-baseline | 83.24 | 64.90 | 54.77 | 61.17 | 58.79 | 65.91 | 64.89 | 61.49 | 68.07 | 70.77 | 62.33 | 69.12 | 70.37 | 77.97 | 68.70 | 52.37 | 63.07 | 54.96 | 52.90 | 51.20 | 54.22 | 53.11 | 51.43 | 51.79 | 50.18 |
| | Shared $Q_{cross}$ | 85.08 | 76.80 | 64.73 | 78.00 | 69.59 | 74.68 | 70.15 | 72.69 | 74.84 | 67.62 | 65.57 | 65.42 | 65.92 | 80.81 | 67.91 | 56.66 | 55.56 | 64.96 | 59.97 | 52.29 | 60.00 | 61.53 | 66.69 | 55.06 | 52.63 |
| | Pair $Q_{cross}$ | 84.84 | 77.38 | 65.67 | 78.04 | 68.24 | 76.45 | 72.22 | 72.99 | 76.16 | 71.22 | 64.66 | 68.31 | 66.13 | 78.86 | 69.48 | 58.05 | 61.33 | 63.40 | 55.63 | 53.20 | 63.21 | 62.44 | 69.03 | 62.44 | 58.61 |
| mix(en, en-zh) | Original | 79.05 | 71.29 | 57.96 | 71.68 | 64.44 | 69.45 | 63.51 | 67.39 | 69.35 | 57.14 | 55.70 | 54.79 | 64.75 | 53.20 | 54.66 | 51.30 | 51.70 | 55.16 | 53.47 | 50.98 | 52.86 | 53.02 | 51.09 | 52.32 | 50.84 |
| | CS-baseline | 83.70 | 67.30 | 58.05 | 62.12 | 61.68 | 67.60 | 62.55 | 66.57 | 67.08 | 68.60 | 60.72 | 70.85 | 83.02 | 63.75 | 67.37 | 50.97 | 60.54 | 56.72 | 54.67 | 51.10 | 54.37 | 52.12 | 51.26 | 53.20 | 50.61 |
| | Shared $Q_{cross}$ | 85.13 | 75.37 | 58.97 | 76.31 | 68.68 | 75.67 | 73.88 | 71.55 | 75.24 | 66.40 | 60.59 | 59.63 | 84.61 | 60.59 | 66.53 | 52.54 | 59.63 | 62.56 | 65.19 | 59.63 | 62.56 | 54.81 | 53.74 | 63.31 | 52.14 |
| | Pair $Q_{cross}$ | 85.84 | 77.68 | 65.88 | 79.35 | 69.83 | 75.51 | 74.44 | 74.03 | 77.83 | 69.78 | 61.06 | 70.41 | 84.41 | 64.64 | 66.73 | 55.98 | 60.18 | 62.16 | 54.81 | 63.94 | 60.29 | 59.37 | 63.00 | 55.12 | |
| **RuleTaker Depth-3** | | | | | | | | | | | | | | | | | | | | | | | | | | |
| mix(en, en-fr) | Original | 73.14 | 64.53 | 56.27 | 65.43 | 58.20 | 63.39 | 57.79 | 60.27 | 63.35 | 69.37 | 63.37 | 62.14 | 53.30 | 54.21 | 58.22 | 50.12 | 52.77 | 57.35 | 52.44 | 48.23 | 52.08 | 52.77 | 48.92 | 48.30 | 48.22 |
| | CS-baseline | 81.14 | 61.44 | 57.71 | 59.62 | 59.98 | 61.67 | 57.72 | 60.66 | 63.21 | 79.67 | 65.99 | 72.52 | 69.99 | 66.44 | 70.13 | 50.20 | 60.14 | 53.56 | 52.25 | 48.58 | 52.37 | 51.54 | 49.79 | 50.28 | 49.19 |
| | Shared $Q_{cross}$ | 81.82 | 71.77 | 62.97 | 74.84 | 64.51 | 72.70 | 65.47 | 68.72 | 71.36 | 75.36 | 69.71 | 68.63 | 59.77 | 60.17 | 67.60 | 53.93 | 55.74 | 70.50 | 59.66 | 48.72 | 57.89 | 60.60 | 50.08 | 50.05 | 48.47 |
| | Pair $Q_{cross}$ | 81.22 | 75.74 | 62.89 | 76 | 65.53 | 73.87 | 67.54 | 68.94 | 70.79 | 78.80 | 69.84 | 73.45 | 65.95 | 62.56 | 72.56 | 52.56 | 57.29 | 56.10 | 55.23 | | | | | | |
| mix(en, en-de) | Original | 75.47 | 67.03 | 57.36 | 64.77 | 59.18 | 64.51 | 58.97 | 59.01 | 64.73 | 61.33 | 59.46 | 58.37 | 51.54 | 55.37 | 69.15 | 51.58 | 52.04 | 54.34 | 56.81 | 48.16 | 51.11 | 51.19 | 48.69 | 48.18 | 48.22 |
| | CS-baseline | 81.01 | 62.66 | 56.65 | 59.06 | 56.94 | 62.67 | 57.49 | 59.36 | 64.22 | 71.86 | 60.81 | 71.01 | 71.03 | 67.64 | 77.44 | 48.99 | 60.67 | 53.75 | 54.36 | 48.05 | 51.36 | 49.91 | 49.94 | 49.75 | 48.72 |
| | Shared $Q_{cross}$ | 83.58 | 73.82 | 60.32 | 77.01 | 65.60 | 74.22 | 68.51 | 70.46 | 74.33 | 66.71 | 63.36 | 63.99 | 75.48 | 82.28 | 51.31 | 53.02 | 60.23 | 51 | 52.10 | 49.85 | | | | | |
| | Pair $Q_{cross}$ | 81.95 | 72.47 | 63.94 | 75.20 | 64.51 | 71.37 | 65.95 | 67.73 | 71.95 | 72.12 | 63.61 | 70.71 | 67.12 | 68.52 | 78.61 | 55.66 | 61.49 | 69.38 | 71.57 | 52.27 | 63.10 | 63.22 | 62.47 | 60.04 | 52.85 |
| mix(en, en-ru) | Original | 74.49 | 67.45 | 61.34 | 68.13 | 62.40 | 65.30 | 62.21 | 62.93 | 66.17 | 57.80 | 58.01 | 56.46 | 58.81 | 65.58 | 54.40 | 55.42 | 55.12 | 52.98 | 51.46 | 48.32 | 51.05 | 50.56 | 49.37 | 48.40 | 48.24 |
| | CS-baseline | 65.79 | 61.80 | 52.81 | 60.92 | 57.93 | 61.38 | 58.66 | 60.99 | 60.26 | 61.52 | 55.15 | 61.01 | 61.12 | 64.49 | 61.54 | 48.30 | 58.33 | 49.97 | 48.48 | 48.11 | 48.87 | 49.01 | 48.48 | 48.37 | 48.03 |
| | Shared $Q_{cross}$ | 83.35 | 72.44 | 63.46 | 76.06 | 65.55 | 74.41 | 68.12 | 70 | 73.58 | 65.23 | 62.01 | 62.01 | 63.69 | 82.03 | 64.05 | 55.19 | 53.81 | 61.10 | 55.85 | 52.83 | 57.10 | 58.32 | 63 | 54.35 | 54.94 |
| | Pair $Q_{cross}$ | 83.64 | 72.77 | 58.50 | 75.21 | 67.41 | 71.66 | 68.61 | 68.20 | 72.81 | 70.80 | 62 | 68.23 | 65.96 | 82.53 | 64.64 | 55.02 | 54.62 | 62.01 | 60.30 | 52.96 | 60.45 | 58.94 | 67.72 | 62.29 | 56.13 |
| mix(en, en-zh) | Original | 74.49 | 66.84 | 57.26 | 66.20 | 61.11 | 66.34 | 62.06 | 64.79 | 66.19 | 57.62 | 57.40 | 55.65 | 70.57 | 53.47 | 54.10 | 52.10 | 51.45 | 54.54 | 51.40 | 48.49 | 51.34 | 51.60 | 49.16 | 52 | 48.31 |
| | CS-baseline | 81.62 | 59.50 | 55.35 | 55.86 | 57.82 | 60.25 | 56.39 | 57.14 | 64.58 | 68.44 | 55.83 | 67.68 | 80.20 | 63.26 | 65.54 | 49.85 | 58.33 | 56.64 | 53.95 | 48.85 | 54.85 | 51 | 50.51 | 50.77 | 50.12 |
| | Shared $Q_{cross}$ | 81.70 | 73.12 | 54.83 | 72.41 | 64.97 | 72.18 | 70.32 | 67.85 | 72.41 | 64.13 | 62.78 | 62.99 | 75.97 | 59.92 | 61.60 | 55.86 | 52.84 | 63.07 | 59.02 | 50.92 | 58.50 | 59.55 | 53.44 | 65.15 | 51.29 |
| | Pair $Q_{cross}$ | 81.97 | 73.41 | 56.51 | 73.12 | 66.06 | 73.97 | 73.47 | 70.10 | 73.43 | 67.25 | 60.47 | 70.28 | 81.64 | 65.12 | 68.34 | 53.91 | 63.81 | 64.99 | 61.65 | 50.22 | 59.70 | 57.81 | 60.30 | 67.36 | 58.34 |

Table 13: Cross-lingual transfer of mBERT model on the **RuleTaker** datasets to either monolingual samples or code-switched language pairs (en-X and X-en). The *original* is the pre-trained model, and the *CS-baseline* is the model that continues pre-training on code-switched data. Shared $Q_{cross}$ and Pair $Q_{cross}$ refer to cases where the cross-lingual query is either shared across many language pairs or is specific to each language pair, respectively. Scores are averaged across three different seeds.

Table 14 — **LeapOfThought (Implicit Evaluation Set)**

| Train Data | Method | en | fr | fa | de | ar | es | zh | ru | it | en-fr | en-it | en-es | en-zh | en-ru | en-de | en-fa | en-ar | fr-en | de-en | fa-en | es-en | it-en | ru-en | zh-en | ar-en |
|---|---|---|---|---|---|---|---|---|---|---|---|---|---|---|---|---|---|---|---|---|---|---|---|---|---|---|
| mix(en, en-fr) | Original | 76.07 | 69.275 | 53.645 | 63.305 | 57.76 | 69.51 | 74.36 | 60.355 | 67.105 | 75.06 | 73.51 | 73.155 | 77.81 | 69.315 | 72.845 | 66.175 | 67.265 | 72.575 | 65.36 | 59.815 | 72.81 | 70.48 | 65.635 | 71.565 | 63.305 |
| | CS-baseline | 71.99 | 67.03 | 50.27 | 64.16 | 57.33 | 65.63 | 67.42 | 56.25 | 59.58 | 70.75 | 66.87 | 69.67 | 74.55 | 64.86 | 68.97 | 56.48 | 66.49 | 51.98 | 68.27 | 58.88 | 59.35 | 66.25 | 62.68 | | |
| | Shared $Q_{cross}$ | 75.80 | 70.29 | 66.25 | 72.30 | 60.28 | 73.55 | 76.49 | 64.55 | 66.95 | 75.33 | 73.62 | 73.70 | 77.58 | 71.45 | 76.11 | 70.91 | 67.42 | 71.92 | 73.39 | 70.52 | 74.24 | 68.97 | 70.99 | 75.17 | 66.41 |
| | Pair $Q_{cross}$ | 79.29 | 72.07 | 53.45 | 66.41 | 60.20 | 72.30 | 77.58 | 62.14 | 68.11 | 78.35 | 76.88 | 77.57 | 80.37 | 73.67 | 78.51 | 68.89 | 72.07 | 74.40 | 74.76 | 61.75 | 74.54 | 72.38 | 68.96 | 75.79 | 66.48 |
| mix(en, en-de) | Original | 77.425 | 71.18 | 55.04 | 71.14 | 59 | 70.05 | 77.54 | 63.925 | 67.145 | 74.83 | 75.525 | 75.68 | 81.23 | 73 | 77.27 | 68.94 | 69.63 | 74.63 | 72.15 | 63.66 | 72.975 | 70.645 | 68.82 | 73.355 | 68 |
| | CS-baseline | 74.24 | 66.41 | 50.97 | 66.80 | 58.57 | 66.87 | 68.04 | 57.41 | 58.34 | 69.43 | 67.18 | 68.74 | 76.49 | 67.88 | 71.99 | 56.87 | 63.69 | 68.43 | 68.43 | 52.99 | 68.97 | 59.19 | 62.53 | 70.36 | 63.07 |
| | Shared $Q_{cross}$ | 77.73 | 69.98 | 57.33 | 71.45 | 58.26 | 70.36 | 73.55 | 62.68 | 66.02 | 75.72 | 75.86 | 75.48 | 80.61 | 74.94 | 77.66 | 71.37 | 72.54 | 72.61 | 68.88 | 74.94 | 71.14 | 69.43 | 75.25 | 68.27 | |
| | Pair $Q_{cross}$ | 79.83 | 70.67 | 57.84 | 71.84 | 60.20 | 71.45 | 76.26 | 65.08 | 68.65 | 78.03 | 75.41 | 77.64 | 81.06 | 75.38 | 80.21 | 69.59 | 72.07 | 75.78 | 74.94 | 64.08 | 76.71 | 71.76 | 71.44 | 77.18 | 69.10 |
| mix(en, en-ru) | Original | 78.17 | 69.98 | 54.66 | 66.91 | 60.01 | 70.09 | 75.84 | 64.43 | 67.03 | 73.29 | 74.72 | 74.68 | 78.99 | 73.40 | 76.03 | 69.72 | 70.83 | 71.73 | 67.30 | 59 | 73.74 | 69.17 | 66.06 | 70.60 | 65.48 |
| | CS-baseline | 72.54 | 66.95 | 50.66 | 64.62 | 57.64 | 65.79 | 64.47 | 57.49 | 61.21 | 70.52 | 67.73 | 69.51 | 75.64 | 67.11 | 69.05 | 60.20 | 64.47 | 66.10 | 65.17 | 52.37 | 66.64 | 63.15 | 59.58 | 64.55 | 62.30 |
| | Shared $Q_{cross}$ | 79.44 | 68.81 | 56.63 | 68.11 | 60.12 | 70.75 | 73.93 | 64.62 | 68.19 | 74.86 | 75.41 | 75.02 | 78.67 | 74.17 | 76.03 | 71.61 | 70.91 | 72.46 | 70.91 | 63.23 | 73.47 | 70.13 | 68.66 | 72.38 | 67.57 |
| | Pair $Q_{cross}$ | 79.98 | 69.51 | 57.64 | 68.96 | 61.20 | 69.82 | 75.71 | 66.18 | 68.11 | 76.64 | 76.80 | 76.33 | 82.29 | 76.33 | 78.34 | 71.45 | 73.14 | 73.31 | 72.45 | 62.53 | 74.70 | 69.59 | 70.26 | 74.01 | 79.03 |
| mix(en, en-zh) | Original | 76.11 | 69.94 | 56.75 | 69.63 | 58.23 | 70.41 | 79.40 | 63.62 | 67.89 | 73.54 | 74.71 | 74.25 | 81.81 | 71.92 | 74.79 | 69.71 | 68.24 | 72.85 | 71.37 | 65.91 | 74.09 | 71.07 | 68.78 | 73.36 | 66.72 |
| | CS-baseline | 74.17 | 66.80 | 50.66 | 67.03 | 60.28 | 66.95 | 74.40 | 58.03 | 59.97 | 70.83 | 67.42 | 68.97 | 80.76 | 66.72 | 69.90 | 55.78 | 62.37 | 67.96 | 67.88 | 52.91 | 69.74 | 62.30 | 63.23 | 69.82 | 63.07 |
| | Shared $Q_{cross}$ | 79.05 | 71.13 | 57.01 | 69.81 | 58.96 | 71.14 | 81.85 | 64.53 | 69.20 | 74.48 | 73.55 | 74.48 | 84.10 | 75.17 | 76.88 | 69.90 | 70.44 | 74.09 | 72.92 | 64.86 | 75.64 | 72.30 | 69.28 | 76.65 | 65.87 |
| | Pair $Q_{cross}$ | 79.29 | 71.68 | 57.18 | 69.67 | 58.81 | 71.14 | 80.76 | 64.16 | 69.05 | 76.95 | 75.72 | 76.26 | 83.86 | 76.63 | 77.19 | 71.92 | 72.52 | 74.16 | 74.63 | 68.27 | 75.25 | 73.47 | 70.67 | 79.13 | 68.10 |

Table 14: Cross-lingual transfer of mBERT model on the **LeapOfThought** dataset to either monolingual samples or code-switched language pairs (en-X and X-en). The *original* is the pre-trained model, and the *CS-baseline* is the model that continues pre-training on code-switched data. Shared $Q_{cross}$ and Pair $Q_{cross}$ refer to cases where the cross-lingual query is either shared across many language pairs or is specific to each language pair, respectively. Scores are averaged across three different seeds.

Table 15 — **RuleTaker** (XLM-R)

| Train Data | Method | en | fr | fa | de | ar | es | zh | ru | it | en-fr | en-it | en-es | en-zh | en-ru | en-de | en-fa | en-ar | fr-en | de-en | fa-en | es-en | it-en | ru-en | zh-en | ar-en |
|---|---|---|---|---|---|---|---|---|---|---|---|---|---|---|---|---|---|---|---|---|---|---|---|---|---|---|
| | | | | | | | | **RuleTaker Depth-0** | | | | | | | | | | | | | | | | | | |
| mix(en, en-fr) | Original | 100.00 | 96.57 | 93.87 | 95.98 | 93.80 | 96.57 | 91.77 | 94.48 | 95.51 | 99.29 | 77.05 | 73.86 | 57.99 | 65.50 | 71.36 | 55.33 | 55.08 | 85.27 | 65.58 | 54.63 | 64.27 | 69.36 | 57.92 | 62.21 | 53.50 |
| | CS-baseline | 100.00 | 93.32 | 90.70 | 94.61 | 93.04 | 96.86 | 94.61 | 94.05 | 96.79 | 99.33 | 76.35 | 75.14 | 61.84 | 66.95 | 71.53 | 58.94 | 58.14 | 79.78 | 64.70 | 52.09 | 66.04 | 68 | 56.33 | 52.48 | 51.89 |
| | Shared $Q_{cross}$ | 100.00 | 96.42 | 95.37 | 96.53 | 93.64 | 96.68 | 93.89 | 94.71 | 96.05 | 99.29 | 80.06 | 79.25 | 69.19 | 70.74 | 76.12 | 62.24 | 61.33 | 92.68 | 73.47 | 59.20 | 72.29 | 77 | 64.03 | 69.99 | 58.30 |
| | Pair $Q_{cross}$ | 99.96 | 96.81 | 95.57 | 95.66 | 94.75 | 96.91 | 92.51 | 94.46 | 96.81 | 98.13 | 86.75 | 80.18 | 74.23 | 76.96 | 84.63 | 60.90 | 65.09 | 90.94 | 82.36 | 64.07 | 76.64 | 83.23 | 74.23 | 73.95 | 60.98 |
| mix(en, en-de) | Original | 100.00 | 95.675 | 91.81 | 97.23 | 94.09 | 97.81 | 87.71 | 94.12 | 96.125 | 69.46 | 68.65 | 67.925 | 52.81 | 59.435 | 99.315 | 56.05 | 52.075 | 69.485 | 87.595 | 56.475 | 65.41 | 67.935 | 58.775 | 60.605 | 53.255 |
| | CS-baseline | 100.00 | 93.14 | 87.09 | 90.83 | 90.43 | 95.18 | 88.75 | 89.96 | 92.89 | 75.84 | 78.59 | 75.58 | 61.01 | 70.85 | 99.67 | 60.26 | 58.45 | 63.38 | 69.38 | 50.35 | 59.67 | 61.26 | 51.58 | 51.43 | 50.03 |
| | Shared $Q_{cross}$ | 100.00 | 95.62 | 95.59 | 97.20 | 94.21 | 98.55 | 93.37 | 95.06 | 96.49 | 78.03 | 77.11 | 74.07 | 70.70 | 66.23 | 99.18 | 59.07 | 56.89 | 76.12 | 95.30 | 60.03 | 70.86 | 74.06 | 64.06 | 71.19 | 55.96 |
| | Pair $Q_{cross}$ | 100.00 | 95.75 | 96.23 | 96.98 | 94.50 | 97.61 | 94.53 | 94.69 | 95.41 | 80.37 | 76.38 | 74.51 | 65.46 | 71.15 | 99.11 | 56.76 | 65.12 | 75.09 | 95.39 | 59.91 | 67.89 | 71.82 | 66.59 | 71.07 | 59.34 |
| mix(en, en-ru) | Original | 99.99 | 96.10 | 91.67 | 94.75 | 91.52 | 97.07 | 89.85 | 93.54 | 95.66 | 71.25 | 72 | 70.97 | 65.49 | 99.42 | 72.23 | 65.25 | 62.03 | 72.23 | 69.88 | 58.06 | 65.09 | 69.05 | 79.66 | 56.84 | 54.33 |
| | CS-baseline | 100.00 | 96.59 | 91.51 | 94.87 | 92.69 | 96.73 | 90.02 | 96.70 | 96.09 | 73.85 | 73.63 | 73.35 | 69.38 | 99.95 | 76.99 | 65.76 | 64.11 | 67.64 | 64.42 | 51.93 | 65.58 | 63.35 | 64.14 | 54.36 | 51.96 |
| | Shared $Q_{cross}$ | 99.78 | 96.53 | 94.52 | 94.64 | 93.07 | 96.58 | 94.53 | 94.40 | 94.86 | 84.03 | 82.06 | 76.87 | 76.90 | 95.56 | 82.07 | 75.43 | 68.67 | 82.55 | 80.08 | 73.04 | 75.93 | 82.69 | 89.68 | 72.86 | 61.02 |
| | Pair $Q_{cross}$ | 99.59 | 96.38 | 94.82 | 94.82 | 93.44 | 95.52 | 94.35 | 94.35 | 94.86 | 85.65 | 83.11 | 79.75 | 81.36 | 95.56 | 84.30 | 74.10 | 72.70 | 82.62 | 80.39 | 74.57 | 84.57 | 76.86 | 68.57 | | |
| mix(en, en-zh) | Original | 100.00 | 95.46 | 94.14 | 95.47 | 94.40 | 97.97 | 94.21 | 92.44 | 96.41 | 70.67 | 70.52 | 71.59 | 99.93 | 65.98 | 67.12 | 63.14 | 60.10 | 72.98 | 70.16 | 57.39 | 65.92 | 69.84 | 64.22 | 71.69 | 54.17 |
| | CS-baseline | 100.00 | 95.49 | 91.12 | 95.19 | 91.27 | 95.76 | 92.35 | 92.86 | 96.02 | 72.14 | 73.87 | 75.24 | 100.00 | 74.34 | 70.51 | 63.83 | 60.11 | 67.54 | 61.89 | 51.83 | 64.17 | 61.02 | 51.90 | 53.25 | 51.77 |
| | Shared $Q_{cross}$ | 99.99 | 96.05 | 96.09 | 96.40 | 94.59 | 98.12 | 97.66 | 93.85 | 96.28 | 76.61 | 79.70 | 76.44 | 99.93 | 73.81 | 78.80 | 68.51 | 63.02 | 76.94 | 79.94 | 65.31 | 74.33 | 74.85 | 73.67 | 93.51 | 60.14 |
| | Pair $Q_{cross}$ | 100.00 | 96.30 | 94.85 | 96.15 | 93.65 | 98.94 | 98.77 | 95.13 | 96.57 | 76.70 | 73.44 | 74.26 | 100.00 | 75.55 | 75.04 | 61.35 | 63.63 | 71.95 | 71.12 | 62.94 | 70.10 | 71 | 73.38 | 98.27 | 60.37 |
| | | | | | | | | **RuleTaker Depth-1** | | | | | | | | | | | | | | | | | | |
| mix(en, en-fr) | Original | 86.26 | 81.37 | 74.45 | 83.07 | 75.39 | 80.86 | 75.84 | 79.84 | 81.59 | 85.79 | 70.28 | 67.60 | 58.43 | 61.52 | 65.83 | 56.53 | 56.82 | 68.40 | 63.25 | 56.38 | 61.91 | 62.31 | 56.01 | 54.58 | 53.59 |
| | CS-baseline | 89.75 | 81.53 | 74.41 | 84.06 | 77.33 | 83.76 | 75.61 | 80.78 | 82.77 | 88.94 | 70.28 | 72.47 | 59.18 | 61.85 | 68.60 | 56.74 | 58.60 | 65.95 | 57.15 | 53.60 | 59.95 | 59.24 | 54.17 | 53.60 | 53.60 |
| | Shared $Q_{cross}$ | 85.65 | 80.88 | 72.46 | 82.58 | 78.34 | 81.32 | 74.38 | 80.12 | 82.68 | 83.55 | 74.88 | 70.59 | 62.14 | 66.91 | 69.58 | 60.06 | 59.96 | 76.52 | 67.48 | 57.93 | 65.99 | 67.02 | 60.25 | 60.16 | 55.40 |
| | Pair $Q_{cross}$ | 88.77 | 84.19 | 79.43 | 84.65 | 80.64 | 86.08 | 81.27 | 82.82 | 84.89 | 89.50 | 74.36 | 73.31 | 68.27 | 66.81 | 70.92 | 57.69 | 60.49 | 79.63 | 67.12 | 52.49 | 66.36 | 60.76 | 63.94 | 65.09 | 56.09 |
| mix(en, en-de) | Original | 87.10 | 84.97 | 80.21 | 83.85 | 80.68 | 83.95 | 78.26 | 81.56 | 82.64 | 67.44 | 68.75 | 65.59 | 54.81 | 59.95 | 90.36 | 57.15 | 56.05 | 77.69 | 55.35 | 62.39 | 55.24 | 57.31 | 58.08 | 54.72 | |
| | CS-baseline | 86.96 | 77.69 | 71.475 | 77.245 | 73.25 | 81.36 | 68.47 | 74.13 | 79.685 | 69.885 | 68.79 | 68.89 | 57.49 | 63.52 | 85.735 | 59.875 | 58.945 | 58.525 | 56.295 | 51.905 | 55.875 | 55.88 | 51.855 | 52.095 | 51.86 |
| | Shared $Q_{cross}$ | 85.66 | 84.10 | 80.64 | 83.35 | 78.74 | 83.43 | 78.91 | 79.95 | 82.73 | 70.99 | 70.36 | 68.67 | 60.30 | 65.37 | 81.43 | 61.58 | 59.29 | 67.50 | 73.86 | 59.50 | 65.77 | 56.53 | 62 | 57.58 | |
| | Pair $Q_{cross}$ | 91.57 | 84.58 | 80.47 | 87.13 | 80.73 | 86.64 | 80.11 | 82.21 | 85.56 | 71.08 | 69.78 | 68.01 | 62.23 | 65.34 | 90.81 | 59.85 | 60.12 | 68.42 | 83.53 | 58.81 | 65.34 | 65.13 | 60.10 | 64.09 | 56.38 |
| mix(en, en-ru) | Original | 86.80 | 80.81 | 77.13 | 82.52 | 77.83 | 82.23 | 77.41 | 81.18 | 81.35 | 66.51 | 67.85 | 65.37 | 62.42 | 84.05 | 65.28 | 61.62 | 50.11 | 64.60 | 61.68 | 55.40 | 61.71 | 62.36 | 59.84 | 56.43 | 54.27 |
| | CS-baseline | 88.70 | 80.39 | 79.94 | 80.89 | 78.92 | 79.27 | 79.82 | 80.18 | 80.50 | 70.08 | 70.51 | 69.10 | 68.21 | 93.39 | 70.53 | 67.03 | 63.37 | 58.51 | 54.40 | 53.44 | 57.05 | 55.20 | 53.39 | 53.50 | 53.32 |
| | Shared $Q_{cross}$ | 91.75 | 83.46 | 79.78 | 84.685 | 78.805 | 85.73 | 80.50 | 84.49 | 84.345 | 74.095 | 73.435 | 70.97 | 68.485 | 89.42 | 71.785 | 65.37 | 63.685 | 71.495 | 68.715 | 61.835 | 68.905 | 69.675 | 80.58 | 65.45 | 57.535 |
| | Pair $Q_{cross}$ | 96.86 | 85.69 | 81.96 | 88.74 | 80.36 | 88.90 | 83.12 | 87.52 | 86.60 | 73.23 | 70.34 | 72.47 | 72.25 | 95.88 | 77.72 | 67.04 | 63.40 | 66.39 | 66.49 | 56.66 | 65.43 | 62.65 | 78.24 | 62.72 | 55.61 |
| mix(en, en-zh) | Original | 86.82 | 82.69 | 78.80 | 83.48 | 80.50 | 84.13 | 79.70 | 81.40 | 83.12 | 64.87 | 66.98 | 62.69 | 84.78 | 62.21 | 64.23 | 59.35 | 58.40 | 65.55 | 63.18 | 56.02 | 64.12 | 64.91 | 60.13 | 56.17 | 54.17 |
| | CS-baseline | 87.61 | 81.06 | 78.53 | 82.79 | 79.06 | 84.08 | 79.46 | 81.62 | 82.53 | 68.43 | 67.75 | 66.80 | 95.41 | 72 | 67.84 | 64.06 | 60.21 | 66.71 | 64.82 | 56.98 | 63.98 | 64.70 | 62.14 | 57.26 | 54.23 |
| | Shared $Q_{cross}$ | 95.93 | 85.09 | 81.10 | 86.16 | 80.62 | 87.49 | 76.19 | 83.15 | 84.97 | 70.38 | 74.47 | 71.31 | 95.41 | 72 | 67.84 | 64.06 | 60.21 | 66.71 | 64.82 | 56.98 | 63.98 | 64.70 | 62.14 | 57.26 | 54.23 |
| | Pair $Q_{cross}$ | 97.25 | 87.50 | 82.49 | 88.02 | 81.33 | 90.99 | 86.15 | 85.81 | 88.41 | 70.42 | 71.53 | 70.30 | 96.41 | 70.59 | 70.54 | 63.84 | 60.97 | 67.66 | 66.18 | 57.65 | 64.13 | 63.09 | 64.57 | 64.73 | 55.02 |
| | | | | | | | | **RuleTaker Depth-2** | | | | | | | | | | | | | | | | | | |
| mix(en, en-fr) | Original | 84.74 | 78.41 | 65.01 | 80.98 | 74.34 | 80.06 | 69.05 | 75.33 | 79.15 | 82.67 | 69.82 | 63 | 52.30 | 61.02 | 64.13 | 51.99 | 52.68 | 67.62 | 61.12 | 54.60 | 61.73 | 62.64 | 56.33 | 54.85 | 52.15 |
| | CS-baseline | 84.67 | 78.76 | 69.39 | 78.70 | 71.20 | 77.10 | 70.10 | 73.95 | 76.14 | 85.48 | 70.14 | 72.59 | 55.91 | 61.47 | 64.51 | 56.31 | 56.11 | 57.07 | 51.77 | 50.74 | 53.32 | 53.52 | 50.74 | 50.74 | 50.74 |
| | Shared $Q_{cross}$ | 93.85 | 81.66 | 73.36 | 82.76 | 72.98 | 82.65 | 74.77 | 77.05 | 80.82 | 92.61 | 74.11 | 75.14 | 63.22 | 67.84 | 71.10 | 60.94 | 60.21 | 75.81 | 64.34 | 55.47 | 64.83 | 65.22 | 59.68 | 58.57 | 53.23 |
| | Pair $Q_{cross}$ | 93.55 | 83.73 | 74.63 | 83.66 | 74.13 | 83.80 | 78.22 | 78.88 | 81.92 | 92.24 | 75.10 | 74.50 | 67.21 | 71.63 | 74.04 | 56.40 | 61.57 | 78.75 | 66.43 | 54.88 | 66.35 | 65.41 | 64 | 63.36 | 55.77 |
| mix(en, en-de) | Original | 86.29 | 78.51 | 75.38 | 82.49 | 74.68 | 79.87 | 75.38 | 76.33 | 76.33 | 65.44 | 64.88 | 56.82 | 59.73 | 85.26 | 57.21 | 55.85 | 58.90 | 69.17 | 53.40 | 57.31 | 58.02 | 54.47 | 53.49 | 51.41 | |
| | CS-baseline | 86.38 | 75.30 | 66.56 | 73.28 | 68.82 | 79.06 | 63.62 | 69.31 | 76.90 | 69.64 | 67.92 | 68.11 | 55.41 | 62.99 | 85.15 | 57.97 | 57.42 | 55.85 | 52.80 | 50.23 | 54.16 | 53.80 | 50.06 | 50.59 | 50.10 |
| | Shared $Q_{cross}$ | 93.45 | 81.25 | 76.28 | 87.04 | 74.32 | 84.69 | 78.49 | 81.51 | 82.25 | 72.40 | 71.75 | 71.85 | 65.40 | 69.83 | 92.21 | 61.35 | 59.27 | 76.73 | 79.20 | 55.94 | 64.07 | 64.56 | 62.72 | 60.43 | 52.61 |
| | Pair $Q_{cross}$ | 94.57 | 81.94 | 73.59 | 85.07 | 72.93 | 83.24 | 72.96 | 76.06 | 80.67 | 73.78 | 71.25 | 74.09 | 70.31 | 71.89 | 93.01 | 58.95 | 60.50 | 68.57 | 81.38 | 54.71 | 64.64 | 63.04 | 64.03 | 63.25 | 56.74 |
| mix(en, en-ru) | Original | 86.47 | 77.31 | 74.55 | 81.67 | 74.53 | 78.16 | 76.08 | 77.24 | 78.05 | 66.34 | 65.15 | 63.11 | 61.05 | 84.12 | 65.64 | 61.29 | 50.50 | 60.09 | 59.26 | 52.03 | 57.50 | 58.69 | 56.29 | 51.46 | 50.91 |
| | CS-baseline | 85.98 | 77.70 | 74.49 | 80.67 | 73.08 | 80.56 | 74.14 | 77.82 | 80.29 | 66.65 | 67.38 | 66.31 | 64.01 | 85.26 | 66.56 | 62.96 | 61.31 | 57.43 | 54.52 | 50.67 | 55.56 | 52.25 | 50.83 | 50.75 | 50.68 |
| | Shared $Q_{cross}$ | 94.09 | 81.92 | 77.10 | 84.43 | 74.51 | 83.24 | 76.92 | 80.62 | 81.45 | 71.36 | 71.66 | 69.47 | 66.52 | 92.24 | 71.05 | 66.17 | 61.58 | 65.93 | 65.70 | 54.34 | 62.08 | 61.89 | 72.73 | 56.71 | 52.77 |
| | Pair $Q_{cross}$ | 92.77 | 79.62 | 74.66 | 82.47 | 75.53 | 81.12 | 77.84 | 80.71 | 80.04 | 71.32 | 67.89 | 70.44 | 69.82 | 91.30 | 75.28 | 63.83 | 61.50 | 63.31 | 54.90 | 60.35 | 58.62 | 68.35 | 56.49 | 55.58 | |
| mix(en, en-zh) | Original | 85.95 | 77.28 | 72.97 | 79.58 | 72.72 | 78.46 | 72.14 | 75.63 | 77.63 | 60.64 | 62.37 | 60.84 | 73.90 | 62.58 | 60.92 | 59.94 | 57.72 | 56.87 | 56.14 | 51.43 | 54.95 | 55.68 | 54.31 | 50.81 | 50.79 |
| | CS-baseline | 85.91 | 77.01 | 72.91 | 81.05 | 73.63 | 81.35 | 78.01 | 76.34 | 79.40 | 65.41 | 64.35 | 64.21 | 85.16 | 63.99 | 60.38 | 57.28 | 57.91 | 52.67 | 50.77 | 55.26 | 53.06 | 50.76 | 50.80 | 50.74 | |
| | Shared $Q_{cross}$ | 92.94 | 81.30 | 76.44 | 83.16 | 73.80 | 83.68 | 81.67 | 78.46 | 81.02 | 70.92 | 72.72 | 70.42 | 92.17 | 70.80 | 68.22 | 64.68 | 60.56 | 66.66 | 65.96 | 55.45 | 65.46 | 64.45 | 67.74 | 53.89 | |
| | Pair $Q_{cross}$ | 95.05 | 82.10 | 73.61 | 80.67 | 73.34 | 81.42 | 77.73 | 76.63 | 80.25 | 70.59 | 72.10 | 70.54 | 93.42 | 70.40 | 65.96 | 64.34 | 61.29 | 63.49 | 60.75 | 52.55 | 60.27 | 58.66 | 62.58 | 63.44 | 55.25 |
| | | | | | | | | **RuleTaker Depth-3** | | | | | | | | | | | | | | | | | | |
| mix(en, en-fr) | Original | 83.48 | 75.78 | 71.145 | 78.13 | 69.65 | 75.22 | 69.63 | 73.62 | 74.81 | 80.53 | 67.36 | 54.36 | 53.37 | 62.53 | 59.73 | 52.68 | 59.14 | 57.35 | 57.33 | 53.78 | | | | | |
| | CS-baseline | 82.92 | 74.19 | 66.55 | 74.59 | 68 | 74.34 | 65.76 | 72.22 | 72.95 | 81.81 | 67.25 | 68.66 | 56.96 | 62.22 | 64.24 | 53.86 | 55.12 | 58.93 | 50.41 | 48.15 | 53.45 | 53.27 | 48.80 | 48.15 | 48.15 |
| | Shared $Q_{cross}$ | 91.32 | 79.12 | 71.26 | 80.01 | 68.78 | 79.74 | 71.69 | 76.12 | 77.31 | 89.03 | 73.41 | 71.22 | 62.21 | 65.53 | 70.77 | 59.46 | 56.78 | 73.69 | 62.17 | 55.54 | 64.71 | 61.25 | 60.66 | 60.32 | 51.09 |
| | Pair $Q_{cross}$ | 90.38 | 75.47 | 70.37 | 79.74 | 68.80 | 77.06 | 71.76 | 74.72 | 75.93 | 88.36 | 73.07 | 70.03 | 64.83 | 65.07 | 70.40 | 58.58 | 61.97 | 79.32 | 61.57 | 55.19 | 65.70 | 60.70 | 60.32 | 59.19 | 59.56 | 55.10 |
| mix(en, en-de) | Original | 89.19 | 76.20 | 70.35 | 79.87 | 67.43 | 75.22 | 69.63 | 73.13 | 74.50 | 68.25 | 68.12 | 66.14 | 52.35 | 61.80 | 86.85 | 57.47 | 54.90 | 60.92 | 69.10 | 52.52 | 58.16 | 60.38 | 57.34 | 55.91 | 49.85 |
| | CS-baseline | 83.44 | 76 | 66.81 | 76.60 | 68.38 | 78.44 | 69.83 | 74.19 | 77.37 | 64.34 | 64.29 | 64.77 | 57.09 | 59.43 | 82.63 | 56.29 | 54.15 | 56.12 | 56.00 | 48.15 | 54.25 | 54.57 | 49.56 | 48.15 | 49.83 |
| | Shared $Q_{cross}$ | 93.34 | 78.42 | 78.14 | 76.93 | 76.82 | 68.86 | 73.46 | 74.96 | 69.09 | 67.39 | 50.92 | 55.75 | 59.85 | 67.57 | 52.91 | 58.58 | 58.59 | 59.26 | 56.20 | 50.82 | 50.86 | | | | |
| | Pair $Q_{cross}$ | 92.29 | 78.34 | 70.37 | 81.37 | 69.60 | 77.98 | 71.29 | 74.22 | 75.14 | 71.52 | 70.62 | 70.47 | 66.69 | 71.60 | 90.52 | 57.72 | 55.98 | 62.75 | 67.63 | 53.41 | 59.91 | 61.63 | 59.62 | 58.61 | 54.71 |
| mix(en, en-ru) | Original | 83.465 | 74.595 | 69.79 | 77.24 | 70.075 | 75.24 | 70.745 | 73.785 | 74.065 | 63.59 | 63.38 | 61.455 | 61.57 | 74.72 | 61.85 | 59.02 | 58.775 | 57.78 | 56.145 | 50.525 | 54.69 | 54.97 | 53.08 | 50.795 | 48.56 |
| | CS-baseline | 75.93 | 69.25 | 68.94 | 67.24 | 65.17 | 69.92 | 62.51 | 69.75 | 68.59 | 64.98 | 64.59 | 62.14 | 51.51 | 49.55 | 48.16 | 51.71 | 49.63 | 48.37 | 48.33 | 48.15 | | | | | |
| | Shared $Q_{cross}$ | 90.46 | 75.68 | 67.05 | 79.83 | 67.25 | 77.19 | 71.49 | 75.66 | 75.78 | 69.14 | 69.02 | 66.25 | 66.28 | 88.83 | 70.13 | 63.51 | 60.17 | 59.99 | 58.93 | 52.08 | 56.63 | 55.82 | 59.14 | 54.36 | 52.85 |
| | Pair $Q_{cross}$ | 90.91 | 76.82 | 69.41 | 80.25 | 68.89 | 78.74 | 74.85 | 77.25 | 76.19 | 68.92 | 66.18 | 66.58 | 67.16 | 90.01 | 68.42 | 63.31 | 61.13 | 59.69 | 52.52 | 58.51 | 56.86 | 66.09 | 58.72 | 52.77 | |
| mix(en, en-zh) | Original | 83.42 | 76.90 | 72.06 | 77.98 | 71.74 | 77.91 | 73.43 | 73.89 | 76.01 | 61.93 | 61.21 | 59.50 | 78.85 | 60.75 | 57.14 | 58.87 | 55.09 | 59.93 | 56.88 | 49.49 | 54.91 | 56.42 | 52.48 | 48.34 | 48.32 |
| | CS-baseline | 83.30 | 75.63 | 68.52 | 76.45 | 68.45 | 77.21 | 71.97 | 73.63 | 74.33 | 62.42 | 61.97 | 62.17 | 59.96 | 57.90 | 54.81 | 52.51 | 52 | 48.53 | 50.63 | 48.76 | 46.18 | 48.14 | 48.15 | | |
| | Shared $Q_{cross}$ | 92.33 | 76 | 68.30 | 78.48 | 68.07 | 78.93 | 70.89 | 71.89 | 75.58 | 67.30 | 68.67 | 66.47 | 91.54 | 65.65 | 64.65 | 60.57 | 54.49 | 58.86 | 58.81 | 51.43 | 56.62 | 55.87 | 55.55 | 57.55 | 48.48 |
| | Pair $Q_{cross}$ | 83.67 | 75.54 | 73.49 | 77.53 | 72.83 | 76.23 | 74.28 | 74.04 | 76.70 | 68.02 | 69.51 | 68.89 | 82.08 | 69.01 | 67.47 | 64.31 | 60.33 | 65.94 | 62.18 | 56.12 | 64.39 | 60.36 | 62.83 | 64.31 | 55.26 |

Table 15: Cross-lingual transfer of the XLM-R model on the **RuleTaker** datasets to either monolingual samples or code-switched language pairs (en-X and X-en). The *original* is the pre-trained model, and the *CS-baseline* is the model that continues pre-training on code-switched data. Shared $Q_{cross}$ and Pair $Q_{cross}$ refer to cases where the cross-lingual query is either shared across many language pairs or is specific to each language pair, respectively. Scores are averaged across three different seeds.

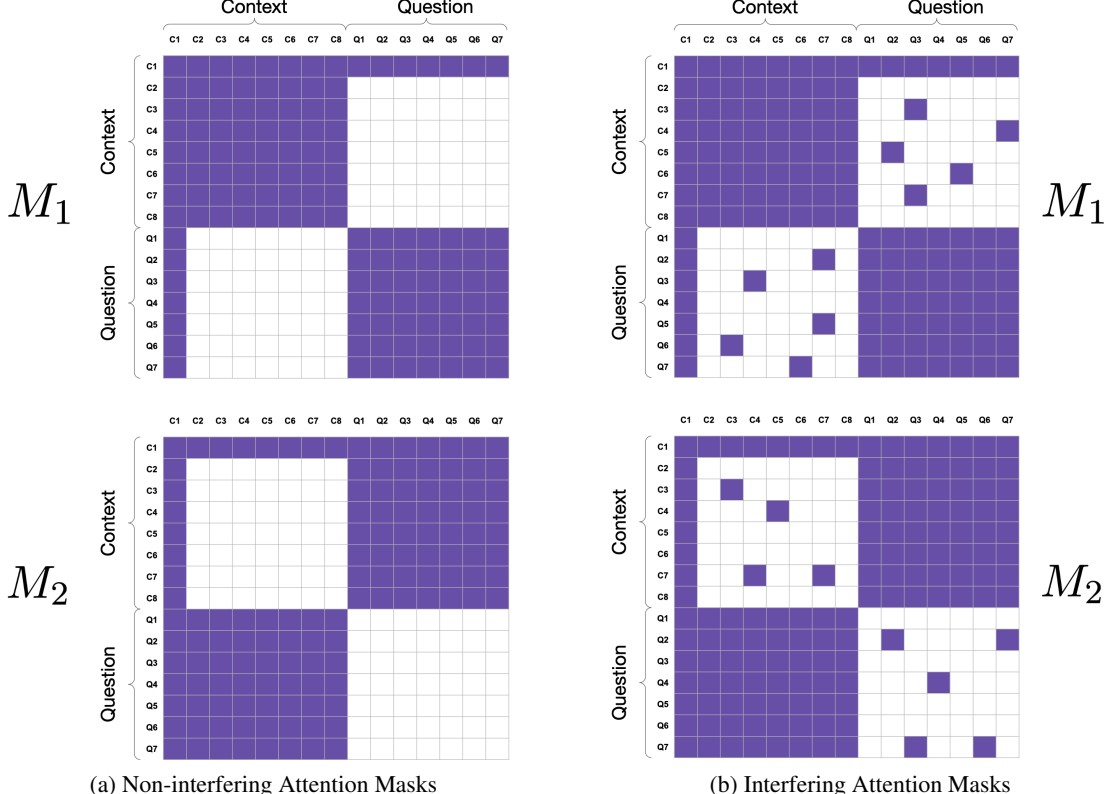

(a) Non-interfering Attention Masks      (b) Interfering Attention Masks

Figure 3: Illustration of the attention masks in Section 4.1. In the proposed scheme, two sets of independent query matrices ($Q$ and $Q_{cross}$) *collaborate* to compute the attention scores. Matrix $M_1$ enforces the $Q$ matrix to mostly focus on monolingual attentions, and matrix $M_2$ constrains the $Q_{cross}$ to mostly handle cross-lingual attentions. The difference between masks in the two figures are the structured attention dropout probability being either one (left) or less than one (right). It is worth noting that the first token (*e.g.*, [CLS] in mBERT) is used as a *bridge* in both $M_1$ and $M_2$, meaning its respective attentions are not masked.

| Train Data | Method | LeapOfThought (Implicit Evaluation Set) | | | | | | | | | | | | | | | | | | | | | | | | |
|---|---|---|---|---|---|---|---|---|---|---|---|---|---|---|---|---|---|---|---|---|---|---|---|---|---|---|
| | | en | fr | fa | de | ar | es | zh | ru | it | en-fr | en-it | en-es | en-zh | en-ru | en-de | en-fa | en-ar | fr-en | de-en | fa-en | es-en | it-en | ru-en | zh-en | ar-en |
| mix(en, en-fr) | Original | 76.50 | 69.94 | 66.22 | 72.87 | 60.92 | 72.85 | 66.73 | 66.22 | 67.15 | 75.57 | 73.40 | 74.14 | 75.81 | 71.77 | 76.31 | 69.67 | 70.48 | 72.07 | 72.54 | 70.76 | 73.86 | 71.34 | 70.33 | 74.09 | 68.62 |
| | CS-baseline | 76.34 | 69.28 | 65.40 | 72.07 | 60.82 | 73.93 | 78.04 | 68.11 | 66.02 | 73.86 | 70.16 | 72.16 | 70.97 | 70.38 | 73.65 | 69.46 | 66.74 | 73.31 | 68.62 | 69.83 | 70.78 | 67.20 | 69.06 | 70.85 | 66.18 |
| | Shared $Q_{cross}$ | 77.11 | 72.23 | 65.63 | 72.15 | 60.90 | 70.67 | 76.96 | 66.72 | 66.49 | 76.80 | 73.86 | 74.09 | 77.81 | 74.24 | 76.73 | 72.77 | 69.82 | 73.31 | 72.07 | 70.83 | 74.17 | 70.21 | 71.76 | 75.33 | 70.05 |
| | Pair $Q_{cross}$ | 77.35 | 72.30 | 67.18 | 74.24 | 63.07 | 74.63 | 78.12 | 66.80 | 66.41 | 74.40 | 74.48 | 73.93 | 78.12 | 73.31 | 76.11 | 72.69 | 70.52 | 74.17 | 74.86 | 72.23 | 75.87 | 72.54 | 69.90 | 75.41 | 69.74 |
| mix(en, en-de) | Original | 76.34 | 67.88 | 67.80 | 71.92 | 61.91 | 73.31 | 78.28 | 66.87 | 65.48 | 72.61 | 72.46 | 72.47 | 76.11 | 69.12 | 75.95 | 70.13 | 65.33 | 66.80 | 72.54 | 71.92 | 74.48 | 70.13 | 70.67 | 75.17 | 70.13 |
| | CS-baseline | 76.42 | 66.02 | 67.42 | 72.61 | 62.53 | 72.69 | 77.27 | 66.56 | 64.70 | 73.32 | 71.54 | 74.17 | 74.05 | 72.46 | 74.26 | 70.92 | 69.90 | 67.73 | 72.46 | 73.31 | 72.69 | 66.80 | 69.12 | 72.54 | 66.87 |
| | Shared $Q_{cross}$ | 76.88 | 66.87 | 66.49 | 73.70 | 61.13 | 72.38 | 77.89 | 67.42 | 67.26 | 73.55 | 72.85 | 74.01 | 77.11 | 71.53 | 76.88 | 71.99 | 67.88 | 68.27 | 74.71 | 71.22 | 74.63 | 69.51 | 72.54 | 77.50 | 68.19 |
| | Pair $Q_{cross}$ | 76.65 | 69.36 | 66.33 | 74.55 | 58.88 | 73.47 | 77.04 | 66.02 | 68.04 | 72.92 | 74.09 | 73.93 | 78.20 | 72.14 | 76.80 | 70.36 | 70.03 | 71.02 | 75.80 | 73.62 | 74.16 | 70.21 | 71.67 | 76.15 | 69.41 |
| mix(en, en-ru) | Original | 76.26 | 69.51 | 69.05 | 72.85 | 60.59 | 73.62 | 76.11 | 67.11 | 67.49 | 71.14 | 71.85 | 72.08 | 74.86 | 71.76 | 74.48 | 70.13 | 66.51 | 66.45 | 71.30 | 70.63 | 74.48 | 70.38 | 70.75 | 76.34 | 66.30 |
| | CS-baseline | 75.04 | 67.18 | 67.34 | 71.69 | 60.68 | 74.86 | 78.98 | 69.05 | 66.18 | 73.70 | 73.31 | 76.11 | 77.97 | 71.32 | 74.66 | 73.55 | 70.21 | 65.89 | 72.17 | 71.99 | 72.17 | 68.66 | 71.76 | 73.34 | 65.49 |
| | Shared $Q_{cross}$ | 76.25 | 68.87 | 69.11 | 71.14 | 59.19 | 75.17 | 77.66 | 67.25 | 66.33 | 73.31 | 72.54 | 74.17 | 76.57 | 73.08 | 75.72 | 71.53 | 68.74 | 68.43 | 72.15 | 71.76 | 76.26 | 70.36 | 71.14 | 77.04 | 68.19 |
| | Pair $Q_{cross}$ | 77.27 | 67.11 | 69.67 | 73.08 | 59.12 | 74.71 | 78.04 | 67.42 | 66.64 | 74.47 | 74.01 | 74.79 | 78.66 | 74.32 | 76.64 | 73.39 | 70.83 | 70.59 | 74.86 | 72.54 | 75.95 | 68.35 | 72.14 | 77.19 | 67.80 |
| mix(en, en-zh) | Original | 75.88 | 68.20 | 66.73 | 72 | 62.45 | 73.93 | 79.99 | 66.33 | 65.02 | 70.24 | 71.93 | 72.78 | 79.60 | 69.80 | 74.71 | 70.61 | 68.52 | 69.74 | 74.65 | 72.23 | 74.94 | 70.67 | 73.31 | 78.43 | 68.50 |
| | CS-baseline | 77.35 | 67.80 | 66.74 | 72.38 | 60.74 | 72.71 | 80.92 | 66.02 | 65.01 | 70.79 | 73.39 | 72.48 | 80.92 | 68.38 | 74.65 | 70.01 | 68.74 | 69.90 | 73.31 | 71.92 | 73.70 | 67.73 | 69.36 | 73.08 | 62.84 |
| | Shared $Q_{cross}$ | 77.66 | 66.25 | 66.18 | 73.31 | 60.05 | 74.63 | 80.45 | 66.80 | 66.49 | 71.37 | 71.84 | 73.16 | 80.61 | 71.37 | 75.48 | 71.99 | 68.19 | 68.19 | 74.24 | 71.53 | 75.95 | 69.90 | 72.61 | 76.65 | 68.19 |
| | Pair $Q_{cross}$ | 77.33 | 67.56 | 66.80 | 73.22 | 60.82 | 73.55 | 79.60 | 66.10 | 67.18 | 71.45 | 72.07 | 73.30 | 80.45 | 71.14 | 76.93 | 72.22 | 68.73 | 70.13 | 74.08 | 71.99 | 74.94 | 70.21 | 72.84 | 76.18 | 69.74 |

Table 16: Cross-lingual transfer of XLM-R model on the **LeapOfThought** dataset to either monolingual samples or code-switched language pairs (en-X and X-en). The *original* is the pre-trained model, and the *CS-baseline* is the model that continues pre-training on code-switched data. Shared $Q_{cross}$ and Pair $Q_{cross}$ refer to cases where the cross-lingual query is either shared across many language pairs or is specific to each language pair, respectively. Scores are averaged across three different seeds.

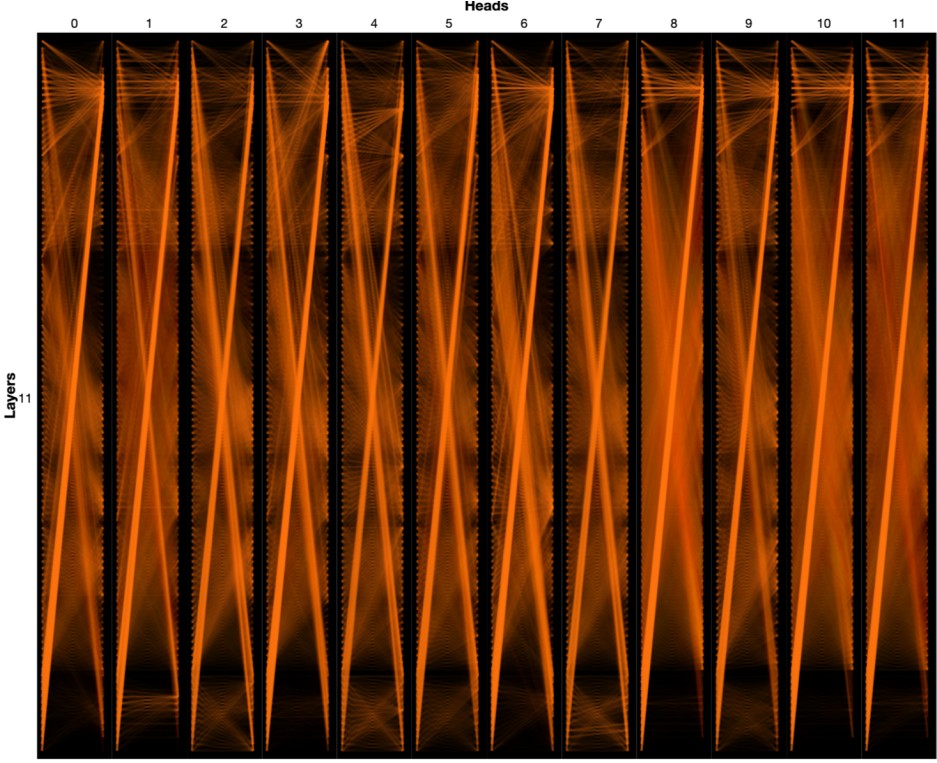

(a) en-fr sample (in-language)

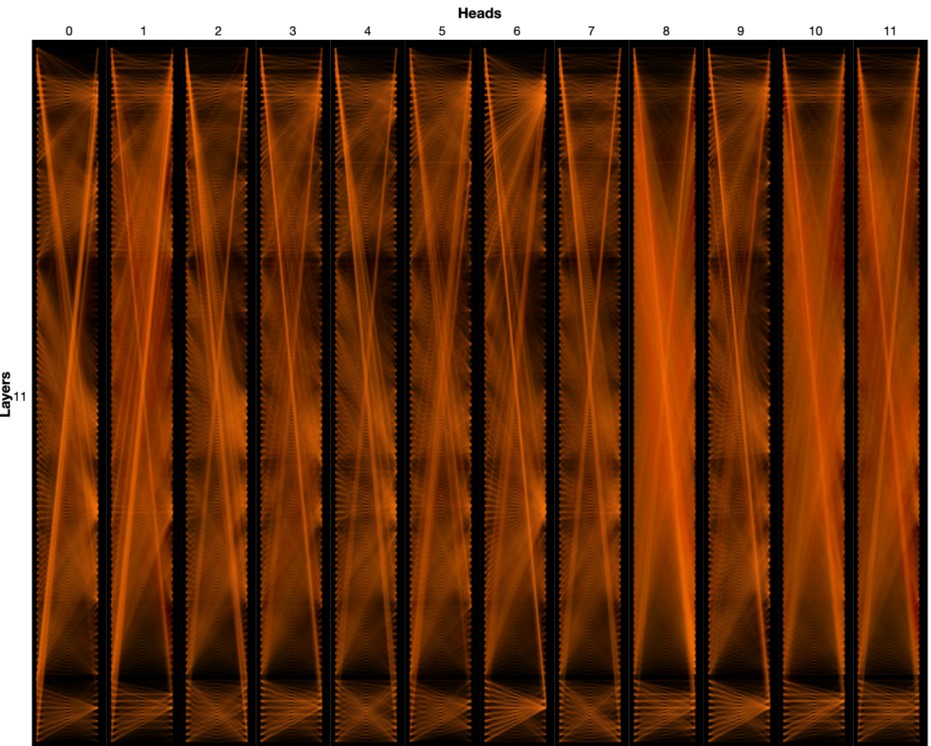

(b) en-ar sample (zero-shot transfer)

Figure 4: Attention visualization of the baseline mBERT model for in-language (en-fr) and zero-shot transfer (en-ar), both from depth-0 of the RuleTaker dataset. The underlying mBERT model is fine-tuned on the mix(en, en-fr) of the RuleTaker depth-0 dataset. We hypothesize that the poor cross-lingual transfer of baseline models to other code-switched languages partially originates from instability of attention patterns across different languages as depicted in above figures.

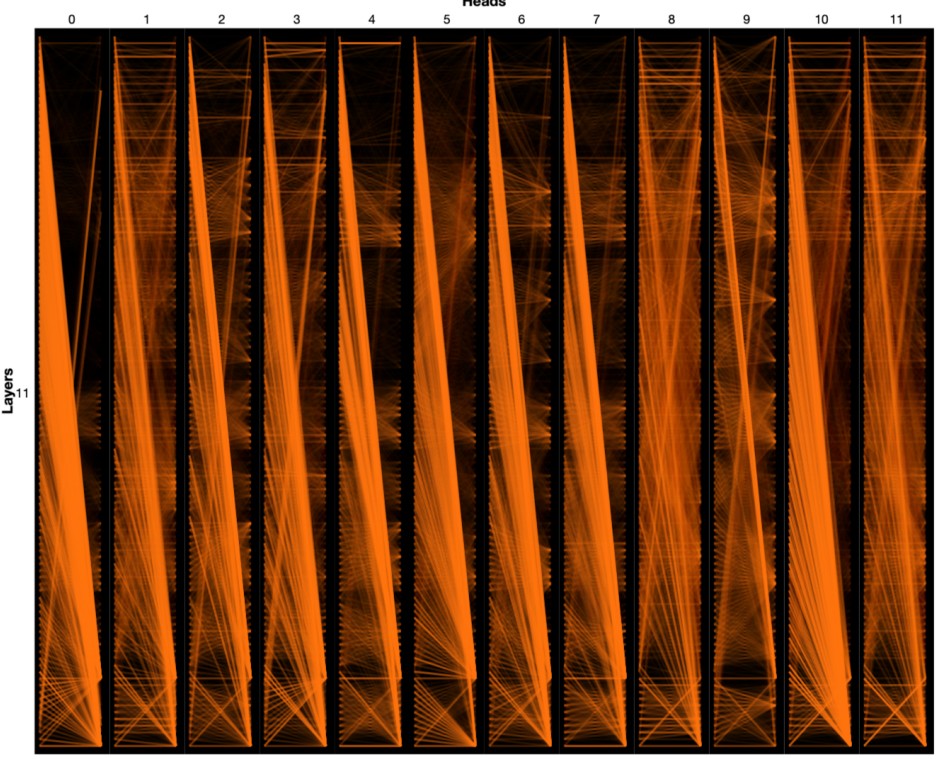

(a) en-fr sample (in-language)

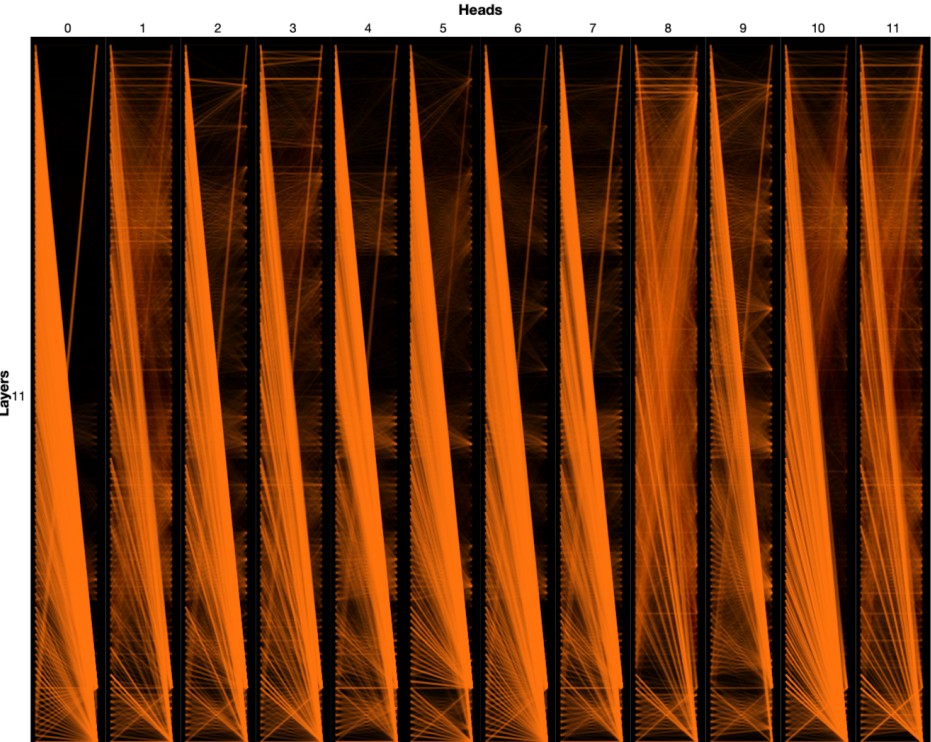

(b) en-ar sample (zero-shot transfer)

Figure 5: Attention visualization of the mBERT model with cross-lingual query for in-language (en-fr) and zero-shot transfer (en-ar), both from depth-0 of the RuleTaker dataset. The underlying mBERT model is fine-tuned on the mix(en, en-fr) of the RuleTaker depth-0 dataset. We can see that attention patterns for our proposed model is more stable between in-language and cross-lingual samples, compared to baseline model in Figure 4.