# OpenReview forum: "Breaking the Language Barrier: Improving Cross-Lingual Reasoning with Structured Self-Attention"
_EMNLP/2023/Conference — EMNLP 2023 Findings_

### Official Review · Reviewer_ocuD · 2023-07-20

**Soundness:** 3

**Excitement:**

3: Ambivalent: It has merits (e.g., it reports state-of-the-art results, the idea is nice), but there are key weaknesses (e.g., it describes incremental work), and it can significantly benefit from another round of revision. However, I won't object to accepting it if my co-reviewers champion it.

**Missing References:**

The paper would benefit from more comprehensive coverage of related works, not only for the purpose of comparison, motivation, and references but also to provide a thorough discussion of the differences between the proposed approach and existing methods. The following one is an example, and many more can be found.

[1] It’s All in the Heads: Using Attention Heads as a Baseline for Cross-Lingual Transfer in Commonsense Reasoning (Tikhonov & Ryabinin, Findings 2021)

**Paper Topic And Main Contributions:**

While previous research has primarily focused on the cross-lingual understanding of multilingual language models (MLMs), the area of monolingual reasoning, particularly in the code-switched setting, has received less attention. This paper aims to fill this gap by addressing the challenge of enhancing cross-lingual transfer, with a specific focus on code-switched alignment, in language models. The key contributions of this paper start with a thorough analysis of MLMs in two distinct scenarios: reasoning over monolingual data and reasoning over code-switched data. The findings from these analyses serve as the foundation for proposing an attention mechanism called cross-lingual aware self-attention. This attention mechanism encourages the model to effectively incorporate cross-lingual attention when processing code-switched sequences. To achieve this, the authors employ techniques such as cross-lingual queries, structured attention dropout, and interfering cross-lingual queries. The experimental evaluation conducted on the RuleTaker and LeapOfThought benchmarks provides evidence supporting the effectiveness of the proposed method. Overall, this paper offers a fresh perspective on challenging the cross-lingual transfer of MLMs in the code-switched setting and presents a corresponding method to mitigate the problem.

**Questions For The Authors:**

A. Based on the experimental results, what is the eventual answer to the research question of whether MLMs can transfer logical reasoning abilities to other languages? Do the results provide conclusive evidence of successful cross-lingual transfer in reasoning tasks, or are there limitations and challenges that need to be addressed?

B. Throughout the discussion of the results, several hypotheses are proposed to explain certain observations (e.g., line 221). Could any evidence or analysis be provided to support each of these hypotheses? Also, how does the lack of code-switched data in pre-training affect the reasoning abilities of MLMs (line 070)? Are any experiments or analyses conducted to validate this hypothesis and establish a stronger connection with the proposed attention mechanism?

C. Without any reference to the previous works and comparison in the same settings, how to verify, for example, whether the implementation and the reported results are correct or not?

D. How to measure the independent contributions of each module of the proposed self-attention mechanism and assess the significance of the improved performance? For instance, can the improvement be primarily attributed to the regularization effects by the dropout over the attention metrics that mitigate potential overfitting issues?

**Reasons To Accept:**

+ The paper breaks new ground to study the reasoning capabilities of MLMs in multilingual settings. By introducing code-switching as a fresh perspective, the paper extends the evaluation of MLMs to more realistic and challenging scenarios, providing valuable insights into how these models handle logical statements in different language contexts, particularly under code-switching.
+ The paper is well-motivated. Firstly, previous works on logical reasoning of language models have mainly focused on monolingual settings, primarily English. Secondly, while there have been studies on the cross-lingual transfer of multilingual language models, they have focused on tasks related to language understanding rather than explicitly investigating the transfer of reasoning abilities. Lastly, the paper highlights that the current multilingual evaluation protocols do not encompass the code-switched setting, which is a common occurrence in real-world language usage.
+ To overcome the challenges of code-switched reasoning, the authors propose an attention mechanism called cross-lingual aware self-attention. By incorporating techniques such as cross-lingual queries, structured attention dropout, and interfering cross-lingual queries, the proposed mechanism aims to encourage the model to effectively utilize cross-lingual information and align it with the given context.
+ The paper conducts thorough experimental evaluations on benchmark datasets, specifically RuleTaker and LeapOfThought. By providing empirical evidence of the effectiveness of the proposed method on these datasets, the paper establishes a benchmark for performance comparison and enables future research to build upon these findings in the direction of code-switched reasoning of MLMs.

**Reasons To Reject:**

+ The paper attempts to lie its contribution in two research goals: evaluating the cross-lingual logical reasoning abilities of MLMs and proposing a method to enhance their multilingual reasoning under coder-switching. However, due to the compressed content, it can be challenging to fully understand and grasp both aspects. For example, although the lack of code-switched data in pre-training is mentioned as a potential reason, there are no subsequent tests or strong connections with the proposed attention mechanism. Meanwhile, the proposed method lacks support from further analysis, and many details can only be found in the Appendix. This limits the clarity and thoroughness of the contributions.
+ The research question is not clearly addressed throughout the paper. While the initial goal is stated as studying the transfer of logical reasoning abilities in MLMs across different languages, the paper does not provide a definitive answer and empirical analysis to this question. Instead, the focus shifts towards proposing a method to improve the performance of MLMs in the code-switched setting. This deviation from the original research question could cause difficulties in following the idea.
+ It appears that the experimental results presented in the paper are not built upon previous work with a formally defined evaluation protocol. The authors introduce both baselines and proposed methods implemented by themselves without any reference to the existing works. This lack of comparison makes it challenging to assess the reliability of the results and the reproducibility of the proposed methods.
+ As mentioned in the first point, there is a lack of clarity on the specific contribution of each module of the proposed cross-lingual-aware self-attention mechanism. While the experimental results indicate that the attention mechanism improves performance, without a detailed analysis or ablation study to isolate the impact of each module, it becomes challenging to understand the proposed methods and assess the significance of the observed improvement.
+ The presentation of the work is not clear. The non-standard organization leads to confusion and hinders understanding. For example, the problem definition is embedded within the motivation section. The analysis of reasoning over monolingual data under the multilingual reasoning section seems out of place and lacks clear justification. The conclusions drawn from the analysis are not explicitly stated, creating a gap between this analysis and the later proposed method. Throughout the experiments, numerous settings involve various language pairs and methods; however, they lack clear and formal definitions and introductions in the first place. As a result, it becomes challenging to follow the flow and understand the statements from these results.

**Reproducibility:**

4: Could mostly reproduce the results, but there may be some variation because of sample variance or minor variations in their interpretation of the protocol or method.

**Reviewer Confidence:**

3: Pretty sure, but there's a chance I missed something. Although I have a good feel for this area in general, I did not carefully check the paper's details, e.g., the math, experimental design, or novelty.

**Typos Grammar Style And Presentation Improvements:**

In general, it is recommended to clearly define the research question, streamline the organization, connect the conclusion from the initial analysis to the introduction of the new method, establish a formal evaluation protocol with references to existing works, conduct additional analysis, and ensure clarity and readability when discussing the experimental results. It may also be worth considering how to strike a balance between showcasing the problem and introducing new methods in a cohesive manner, given limited pages.

---

> ### Author Rebuttal · Authors · 2023-08-29
>
> We thank the reviewer for their comments about our thorough experiments and analysis and our novel perspective on code-switching.
>
> ---
> ### Questions:
>
> >**\[Q_A\]:** The reviewer asks for an eventual answer for the logical reasoning ability of MultiLMs. Our experiments and analysis conclude that multilingual language models can exhibit strong logical reasoning capabilities in monolingual settings and show high cross-lingual transfer performance. However, they struggle when it comes to the code-switched setting. So, the main challenge and limitation is that these models fail to generalize their reasoning ability in a code-switched scenario. The code-switched setting can be commonly found in realistic scenarios where questions and contexts are in different languages, and it is important for multilingual language models to be able to handle such a scenario.
>
> ---
> >**\[Q_B\]:** The reviewer asks for clarification on line 221, regarding potential sources of systemic bias in the LeapOfThought dataset. Our hypothesis is based on the (empirical) observation of artifacts in the Explicit development set of the English LoT dataset (see Appendix A2.1). We hypothesize that some artifacts in the Implicit dev set also exist that might enable the model to learn in-language shortcuts that are not generalizable to other languages.\
> The reviewer also asks for more explanation about why the lack of code-switched data in pre-training affects the model’s performance in our multilingual setting:
> Line 070 highlights MultiLMs’ struggle to reason over code-switched data (i.e., non-monolingual reasoning). These models have not seen code-switched data during pre-training, so they are not optimized for such an input. Hence, when the data format and domain are not consistent for pre-training and fine-tuning, the model is more likely to overfit the fine-tuning data and not transfer to other data formats.
> To demonstrate the effect of adding code-switched data for the pre-training phase, we have a code-switched baseline (**CS-baseline**), in which the MultiLM is pre-trained on a code-switched version of XNLI (continue LM pre-training using a code-switched data). This baseline improved transfer results significantly for en-X format, showing the inclusion of code-switched data in the pre-training phase and making it consistent with the fine-tuning phase plays a role in the model’s performance to reason over code-switched data (Table 3). However, this baseline negatively affects the performance in the monolingual scenario and performs worse than our proposed cross-lingual query matrix method.
>
> ---
> >**\[Q_C\]:** The reviewer is concerned that our method is not compared to results from previous work. As we are the first ones to design an experimental framework for code-switched logical reasoning, previous work has been done only on English data. We are happy to include the English baselines from these work for the final version of the paper to make it easier for the reader to compare. Here are the baseline results (accuracy) for English using our implementation on RoBERTA-Large (used in both RuleTaker and LeapOfThought papers):
> >>|     Model     | RuleTaker [1] | RuleTaker | RuleTaker | RuleTaker | LeapOfThought [2] |
> >>|:-------------:|:-------------:|:---------:|:---------:|:-----------------:|:-----------------:|
> >>|               |    Depth-0    |  Depth-1  |  Depth-2  |  Depth-3  | Implicit dev set  |
> >>| RoBERTA-Large |    100.00     |   99.67   |   99.22   |   98.58   |       88.39       |
> >>|     mBERT     |    100.00     |   93.37   |   88.00   |   88.46   |       81.15       |\
>
> >We note these numbers are higher than our english-language results for mBERT, but that (1) mBERT is based on the BERT-small architecture of ~110M parameters while RoBERTa-large has ~345M parameters, and (2) RoBERTa-Large was trained on far more data. We will publicly release the code and the datasets used in this work, along with the final version of the paper.
>
> ---
> >**\[Q_D\]:** The reviewer asks for the contribution of each component of the proposed method (structured dropout & cross-lingual query).
> We have done an ablation study to evaluate the effect of each component in isolation in Tables 10 and 11 (in Appendix D.1 and D.2) and mention them in the main body (lines 317 and 328 of Section 4). The results show that both cross-lingual query matrix and structured drop attention play roles in the improvement of the model’s performance.
> We will move these results from the appendix to the main body with the extra page we get for the camera-ready version.\
> [1] Clark, Peter, Oyvind Tafjord, and Kyle Richardson. "Transformers as soft reasoners over language." arXiv preprint arXiv:2002.05867 (2020).\
> [2] Talmor, Alon, et al. "Leap-of-thought: Teaching pre-trained models to systematically reason over implicit knowledge." Advances in Neural Information Processing Systems 33 (2020): 20227-20237.
>
> ---
> ### Concern Statements:
> >**\[Concern Statement 1\]:**
> The reviewer is concerned that the paper is compact because we address two research questions. We agree that this is a dense paper; however, because no prior work addressed the cross-lingual logical reasoning in MultiLMs, we had to first evaluate their performance on this task to highlight the problem in the code-switched setting and then propose a method to mitigate the problem. We view highlighting the problem as an important contribution of this paper.
>
> ---
> >**\[Concern Statement 2\]:**
> The reviewer is concerned about the research question not being addressed clearly throughout the paper. In both the Abstract and Introduction, we set two main research questions for the paper: 1) studying the cross-lingual transfer ability of MultiLMs in monolingual and code-switched settings, and 2) proposing a method to improve the cross-lingual performance of the model in logical reasoning. Section 3 is dedicated to answering the first question and highlighting the model’s struggle in a code-switched setting. We propose a solution to mitigate the problem in Section 4, which our results confirm provides an improvement.
>
> ---
> >**\[Concern Statement 5\]:**
> The reviewer has suggestions for the structure of the paper.\
> We thank the reviewer for pointing out what could be clearer. If accepted, we will incorporate some of these suggestions in our revision for the camera-ready.

---

### Official Review · Reviewer_SPqx · 2023-08-04

**Typos Grammar Style And Presentation Improvements:** Line 108
**Soundness:** 4

**Excitement:**

4: Strong: This paper deepens the understanding of some phenomenon or lowers the barriers to an existing research direction.

**Missing References:**

None.

**Paper Topic And Main Contributions:**

The paper examines the cross-lingual reasoning ability of mBERT and XLM-r models. Using the Google Translator API, the authors translate the RuleTaker benchmark and LeapOfThught dataset from English into eight languages, including French, German, Chinese, Russian, Spanish, Farsi, Italian, and Arabic. They evaluate the performance of multilingual models in the cross-lingual zero-shot setting and code-switched format where the context and statement languages are not the same.
The proposed results show that in the monolingual and cross-lingual settings (where the fine-tuning language differs from the evaluation), the multilingual models have high performance on this task. However, as the depth of reasoning increases, the performance drops. The results show a similar trend in the code-switched setting where the fine-tuning and evaluation data have the same languages (e.g., en-fr). Nevertheless, the performance is pretty low when the statement or context language is not English and differs from the fine-tuning language.
To enhance the cross-lingual performance of this setup, the paper suggests a cross-lingual attention approach. In the proposed method, they learn a new cross-lingual query matrix using code-switched unsupervised data. The learned query matrix could be shared between all possible language pairs or be specific for two languages. During the evaluation, they use the cross-lingual query matrix instead of the fine-tuned one. The results show that the proposed method is effective for both monolingual instances and code-switched ones.

**Questions For The Authors:**

I have a few suggestions that could improve the clarity of the paper:
- The reason for using BitFit for fine-tuning models is not clear in the main paper. I would suggest bringing a part of the C1 results to the main paper. This is interesting that BitFit enhances cross-lingual performance.
-  The results of A.2.1 is quite interesting to me. It would be better to have a short mention of this in the main paper.

**Reasons To Accept:**

The paper has a fresh look at analyzing the cross-lingual reasoning ability of multilingual models using code-switched structured instances. Besides, the proposed method is a computationally efficient approach (compared to the pre-training) that can adopt existing multilingual models to the code-switched data. Structurally, the paper is well-written and easy to follow.

**Reasons To Reject:**

Some missing details in the results (or the motivation) make understanding difficult. I've added more specifics in the "Questions for the authors" to clarify the findings.

**Reproducibility:**

4: Could mostly reproduce the results, but there may be some variation because of sample variance or minor variations in their interpretation of the protocol or method.

**Reviewer Confidence:**

4: Quite sure. I tried to check the important points carefully. It's unlikely, though conceivable, that I missed something that should affect my ratings.

---

> ### Author Rebuttal · Authors · 2023-08-29
>
> We appreciate the reviewer’s comment on the novelty of the analysis and the efficiency of the proposed method.
>
> ---
> >**\[Q1, Q2\]**: The reviewer has suggestions regarding the explanation of Bitfit fine-tuning and the artifacts of the LeapOfThought dataset:
> In the main body, we briefly refer to our observation on Bitfit fine-tuning and the artifacts of LeapOfThougth, which are present in the appendix (i.e., C1 and A.2.1). However, as suggested by the reviewer, we are willing to move the related information from the appendix to the main body for the final version.\
> We will add more details on these aspects with the extra page we get for the final version of the paper.

---

### Official Review · Reviewer_Gdvn · 2023-08-04

**Soundness:** 4

**Excitement:**

4: Strong: This paper deepens the understanding of some phenomenon or lowers the barriers to an existing research direction.

**Paper Topic And Main Contributions:**

The paper presents evaluation analysis of multilingual models on monolingual and code-switching language transfer using two logical reasoning datasets. The key finding from the evaluation is that while multilingual models show high zero-shot monolingual transfer (when both context and question of the task are in the same language), they suffer in the code-switching scenario (where context and question are from different languages). The paper then proposes an attention mechanism to train dedicated cross-lingual query matrices. This proposed mechanism shows improvement in the code-switching transfer experiments. The contributions are: (1) evaluation analysis on code-switching transfer for multilingual models (2) proposal of a new attention mechanism that improves multilingual models' performance in code-switching transfer.

**Reasons To Accept:**

- The paper presents an interesting and rigorous analysis on how multilingual models are suffering in the code switching transfer scenario and offers convincing explanations. This analysis is an interesting scientific study on the limits of these multilingual models with regard to cross lingual transfer.

- The proposed methods prove to be very effective as it significantly improved the performance of models on the transfer setting.

**Reasons To Reject:**

- The main issue of the paper is that the experimental details are very unclear and the reader has to guess in many cases. Below are some of such examples:
(1). For the cross-lingual results in Table 1, did the authors fine-tune the model on the source language and test on every other language and then take an average of the results of these target language performance? It would be good to explicitly state this.
(2). What is the mono performance in Table 3? Is it first training from English + de/fr/ru/zh and fine-tune on fr/de/ru/zh, and then zero-shot transfer to the other three unseen languages from fr/de/ru/zh?
(3). In line 370, the author(s) mentioned that only cross lingual query matrix was trained, but why is there monolingual English training data (in this case there is no cross lingual query matrix and all the parameters will be fixed)?

- All the analysis is based on Google translations of the datasets, which introduces additional noise due to potential errors in machine translation. Why not use existing manually written/verified datasets such as XNLI, XCOPA?

- Maybe a limitation here rather than a weakness. The code-switching setup in this study is a little bit artificial with the analysis and the model being specific to the datasets and tasks, while code-switching in natural language tends to switch specific phrases and has specific rules and tendencies. That being said, I think the analysis from the paper is a valid scientific study on the hypothetical language switching behaviours of multilingual models.

**Reproducibility:**

4: Could mostly reproduce the results, but there may be some variation because of sample variance or minor variations in their interpretation of the protocol or method.

**Reviewer Confidence:**

4: Quite sure. I tried to check the important points carefully. It's unlikely, though conceivable, that I missed something that should affect my ratings.

---

> ### Author Rebuttal · Authors · 2023-08-29
>
> We thank the reviewer for highlighting the importance and rigor of our analysis and the effectiveness of the proposed method.
>
> ---
>
> Regarding the clarification questions asked by the reviewer:
>
>
>
>
> >**\[Q1\]**: The reviewer asks for clarification on Table 1, which reports results for when we fine-tune models on a single source language, test it on other languages, and then take an average of these results. The reviewer’s interpretation is correct. We report the average cross-lingual transfer performance of each source language. We will clarify this in the camera-ready.
>
> ---
> >**\[Q2\]**: The reviewer asks for clarification on **“mono”** performance in Table 3, where we report the zero-shot cross-lingual transfer performance of the proposed method and the baselines for two monolingual and code-switched settings. Column **“mono”** shows the zero-shot transfer performance of the model in a **“monolingual”** setting (when context and question are in one language). More precisely, in Table 3, we fine-tune the model on English + fr/de/ru/zh (e.g., mix(en, en-fr)) and zero-shot transfer it to unseen languages (e.g., it, es, fa, ar). The results of this column shows that our approach improves the transfer performance for monolingual reasoning in many languages.
>
> ---
> >**\[Q3\]**: The reviewer asks why we include monolingual data (English) while learning cross-lingual query matrices (line 370). For the **pre-training** phase, we only use code-switched data to learn cross-lingual query matrices while the rest of the parameters are frozen. However, for the **fine-tuning** phase (when we learn the reasoning task), we include English training data to train a multilingual reasoner that can reason over both monolingual and code-switched data. During the inference on monolingual data (e.g., English), the model’s original query matrix is used to compute attention scores, and for the code-switched data, both the original and the cross-lingual query matrices are involved.
>
> ---
> >**\[Concern Statement 2\]**:
> The reviewer is concerned about the translation of the datasets and asks why we did not use manually verified datasets like XNLI and XCOPA:
> We use logical reasoning tasks to study the cross-lingual reasoning ability of multilingual models because they allow us to control the exact degree of code-switched text in the contexts and questions. As there is no multilingual logical reasoning dataset, we automatically translated existing English datasets (i.e., RuleTaker and LeapOfThought).
> However, to show the proposed method’s generality and address the reviewer’s concern, we performed an experiment on the XNLI dataset. We fine-tuned the mBERT model on a combination of EN and code-switched EN and FR data (mix(en, en-fr)), then zero-shot transfer it to other languages for monolingual evaluations (excluding en and fr) and other language pairs for code-switched evaluation (excluding en-fr and fr-en pairs). Finally, we get the average of these zero-shot transfer results (accuracy). We observe ~4%  improvement on en-X,  ~7% on X-en, and competitive performance on monolingual evaluation setups, indicating the effectiveness of our proposed method on downstream tasks other than logical reasoning.
> The languages included in the evaluation are German, Russian, Spanish, Chinese, and Arabic, and the detailed results are:
> >>| Fine-tune Data Language |      Model     |    mono    |  en_X |  X-en |
> >>|:-----------------------:|:--------------:|:----------:|:-----:|:-----:|
> >>|            en           |    Original    | 69.42  [1] | 63.16 | 68.79 |
> >>|      mix(en, en-fr)     |    Original    |    68.35   | 67.43 | 65.18 |
> >>|      mix(en, en-fr)     |   CS-Baseline  |    68.26   | 70.58 | 65.89 |
> >>|      mix(en, en-fr)     | Shared Q_cross |    68.30   | 69.22 | 70.16 |
> >>|      mix(en, en-fr)     |  Pair Q_cross  |    69.31   | 71.53 | 72.40 |
>
> >We will include the complete evaluation results for the XNLI dataset in the final version of the paper. \
> For the XCOPA dataset, unfortunately, the training dataset is only available in English and would need translation for the fine-tuning of code-switched data.\
> **[1]** Hu, Junjie, et al. "Xtreme: A massively multilingual multi-task benchmark for evaluating cross-lingual generalisation." International Conference on Machine Learning. PMLR, 2020
>
> ---
> >**\[Concern Statement 3\]**:
> The reviewer is concerned about the code-switched setting being artificial. Although in this paper, we address the structured code-switched problem in a controlled setup, the code-switched setting can be found in many realistic scenarios, such as when non-English speakers may ask questions about information that is unavailable in their native language. In this case, the model needs to combine the context information in one language with the question in another. That’s why it’s important for the model to have the ability to perform in such a setting.
>
> We will add more explanations in the body and captions of the tables to clarify the reported results.

---

### Meta-Review · Area_Chair_DJ58 · 2023-09-18

**Recommendation:** 3

**Metareview:**

The paper focuses on improving the cross-lingual reasoning of multilingual language models (here XLM-R).

It introduces a learnable query layer (as part of the self-attention layers) trained specifically on code-mixed data to help the model to reason on context in one language to answer a question in another language.
The method leads to better cross-lingual performance compared to baselines.

Reasons to Accept:
- New empirical evidence that adding adapter-like parameters as the query in the self-attention and explicitly trained in the cross-lingual setting helps the cross-lingual reasoning abilities of the MLMs.  (R1, R2, R3)

Reasons to Reject:
- Lack of mention, discussion, and comparison with related methods such as adapters (Pfeiffer et al. 2020)
- Running the experiments exclusively on machine-translated may lead to noisy results (R3).
- Cross-lingual reasoning defined in the paper is a synthetic task that requires further discussion and comparison with existing cross-lingual tasks (R1 and R3)

---

### Decision · Program_Chairs · 2023-10-07

**Decision:**

Accept-Findings

**Comment:**

The paper focuses on improving the cross-lingual reasoning of multilingual language models (here XLM-R).

It introduces a learnable query layer (as part of the self-attention layers) trained specifically on code-mixed data to help the model to reason on context in one language to answer a question in another language.
The method leads to better cross-lingual performance compared to baselines.

Reasons to Accept:
- New empirical evidence that adding adapter-like parameters as the query in the self-attention and explicitly trained in the cross-lingual setting helps the cross-lingual reasoning abilities of the MLMs.  (R1, R2, R3)

Reasons to Reject:
- Lack of mention, discussion, and comparison with related methods such as adapters (Pfeiffer et al. 2020)
- Running the experiments exclusively on machine-translated may lead to noisy results (R3).
- Cross-lingual reasoning defined in the paper is a synthetic task that requires further discussion and comparison with existing cross-lingual tasks (R1 and R3)